# PRISMM-BENCH: A BENCHMARK OF PEER-REVIEW GROUNDED MULTIMODAL INCONSISTENCIES

**Lukas Selch**[1]    **Yufang Hou**[2]    **M. Jehanzeb Mirza**[3]    **Sivan Doveh**[4]
**James Glass**[3]    **Rogerio Feris**[5]    **Wei Lin**[1]

[1]Johannes Kepler University Linz    [2]Interdisciplinary Transformation University Austria
[3]MIT CSAIL    [4]Stanford University    [5] MIT-IBM Watson AI Lab
🌐 https://da-luggas.github.io/prismm-bench/

## ABSTRACT

Large Multimodal Models (LMMs) are increasingly applied to scientific research, yet it remains unclear whether they can reliably understand and reason over the multimodal complexity of papers. A central challenge lies in detecting and resolving inconsistencies across text, figures, tables, and equations, issues that are often subtle, domain-specific, and ultimately undermine clarity, reproducibility, and trust. Existing benchmarks overlook this issue, either isolating single modalities or relying on synthetic errors that fail to capture real-world complexity. We introduce PRISMM-Bench (*P*eer-*R*eview-sourced *I*nconsistency *S*et for *M*ultimodal *M*odels), the first benchmark grounded in real reviewer-flagged inconsistencies in scientific papers. Through a multi-stage pipeline of review mining, LLM-assisted filtering and human verification, we curate 384 inconsistencies from 353 papers. Based on this set, we design three tasks, namely inconsistency identification, remedy and pair matching, which assess a model's capacity to detect, correct, and reason over inconsistencies across different modalities. Furthermore, to address the notorious problem of *choice-only shortcuts* in multiple-choice evaluation, where models exploit answer patterns without truly understanding the question, we further introduce structured JSON-based answer representations that minimize linguistic biases by reducing reliance on superficial stylistic cues. We benchmark 21 leading LMMs, including large open-weight models (GLM-4.5V 106B, InternVL3 78B) and proprietary models (Gemini 2.5 Pro, GPT-5 with high reasoning). Results reveal strikingly low performance (27.8-53.9%), underscoring the challenge of multimodal scientific reasoning and motivating progress towards trustworthy scientific assistants.

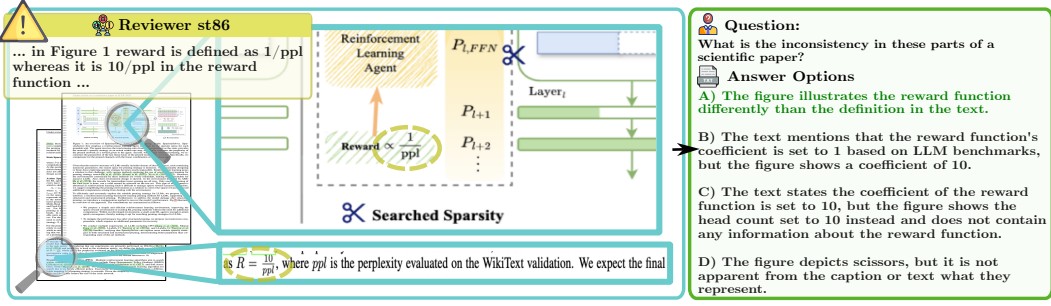

Figure 1: We collect reviewer-flagged inconsistencies in scientific papers and transform them into QA tasks that probe detection, correction, and reasoning over multimodal inconsistencies.

# 1 INTRODUCTION

Recent advances in Large Multimodal Models (LMMs) have sparked growing interest in their potential to serve as intelligent assistants for scientific research (Eger et al., 2025), supporting tasks such as figure and chart interpretation (Roberts et al., 2024; Tanasă & Oprea, 2025; Wu et al., 2024), paper summarization (Tan et al., 2025; Saxena et al., 2025; Yu et al., 2025), and error detection (Yang et al., 2025b; Miyai et al., 2024; Alsaif et al., 2024). Yet, a central open question remains: *can LMMs truly reason over the complex multimodal structure of scientific documents?*

A central challenge in this setting is detecting and resolving inconsistencies between text, figures, tables, or equations in scientific papers. These issues are often subtle, arising from copy-paste mistakes, outdated results, or inconsistent notation, and require domain knowledge to detect. Fig. 1 illustrates such a case, where the reward function is defined differently in the figure and the in-line text. Our analysis of ICLR 2025 submissions reveals that 17.0% contained at least one such inconsistency flagged by reviewers. These discrepancies undermine clarity, reproducibility, and ultimately scientific trust.

Existing benchmarks, however, fall short of exposing this. Document QA datasets (Mathew et al., 2021; 2022; Zhu et al., 2022) or standalone scientific visual element tasks focusing on diagrams, charts, or tables (Kafle et al., 2018; Masry et al., 2022; Cheng et al., 2022) miss the multimodal dependencies of scholarly works. Synthetic datasets (Yan et al., 2025) generate artificial errors, but these are often obvious and unrepresentative of real-world complexity. However, constructing such a benchmark of authentic inconsistencies is challenging as these cases are rare, scattered, and labor-intensive to verify, often requiring domain expertise to identify and validate. To address this, we leverage a valuable but underutilized resource - *open peer reviews*, focusing on instances where reviewers flag mismatches across different modalities, such as between text, figures, tables, and equations, thereby providing a natural source of real-world, human-identified inconsistencies.

In this paper, we introduce PRISMM-Bench, a *P*eer-*R*eview-sourced *I*nconsistency *S*et for *M*ultimodal *M*odels. Unlike synthetic datasets, PRISMM-Bench captures inconsistencies explicitly flagged by human reviewers in scientific papers on OpenReview. Through a multi-stage pipeline combining large-scale review mining, LLM-assisted filtering, and rigorous human verification, we curate a dataset of 384 inconsistencies across 353 papers submitted to ICLR 2024 & 2025, spanning 15 categories of visual-textual and inter-visual mismatches. PRISMM-Bench provides a principled resource for evaluating and improving LMMs, grounded in the real challenges of understanding and verifying scientific papers. Grounded on these recent reviews, we minimize data contamination risk and demonstrate a pipeline with the potential to construct a continuously updated live benchmark. Building on this inconsistency set, we design a benchmark suite of three multiple-choice question (MCQ) tasks: 1) Inconsistency Identification - detect what the inconsistency is; 2) Inconsistency Remedy - determine how to fix the inconsistency; and 3) Inconsistency Pair Match - identify which two elements conflict. Together, these tasks form a tiered framework that evaluates not only a model's ability to spot inconsistencies but also its capacity to propose remedy, and reason over relationships between different modality components.

In MCQ evaluation, a key challenge is models' tendency to rely on linguistic biases in answer choices. Prior work has shown that LLMs often exploit choice-only shortcuts, achieving non-trivial accuracy without reading the question (Chandak et al., 2025; Turner & Kurzeja, 2025; Balepur et al., 2024; Chizhov et al., 2025), and similar effects appear in multimodal MCQs (Chandak et al., 2025). To address this, we propose a novel structured JSON-based answer representation that de-emphasizes stylistic cues and minimizes spurious correlations. Inspired by Das et al. (2014) and Banarescu et al. (2013), our design converts free-form natural language into uniform structured representations that reduce model sensitivity to surface-level patterns. Our user study confirms that this approach suppresses linguistic shortcuts and better aligns models with human reasoning.

We benchmark 21 state-of-the-art LMMs, spanning large open-weight models such as GLM-4.5V 106B (Hong et al., 2025) and InternVL3 78B (Zhu et al., 2025), as well as leading proprietary models including Gemini 2.5 Pro (Comanici et al., 2025) and GPT-5 (OpenAI, 2025). Results show that while large open-weight models achieve around 40% accuracy, even the strongest proprietary models reach just 53.9%, underscoring difficulty of the benchmark and limitations of current LMMs.

Our contributions are fourfold: (1) We propose a reviewer-sourced dataset of real multimodal inconsistencies in scientific papers, spanning diverse categories and grounded in peer review. (2) We construct a benchmark suite of three tasks probing detection, correction, and relational reasoning over these inconsistencies. (3) We are the first to propose JSON-based debiasing method for MCQ, converting free-form responses into uniform structured representations. (4) We evaluate 21 state-of-the-art LMMs, exposing their current limitations in detecting, understanding and correcting inconsistencies in scientific papers.

## 2 RELATED WORK

**Large Multimodal Models (LMMs).** Large Multimodal Models (LMMs) pair a vision encoder with a large language model, enabling open-ended reasoning across tasks such as image captioning, VQA, document understanding, and chart interpretation. Early approaches like BLIP-2 (Li et al., 2023) and InstructBLIP (Dai et al., 2023) introduced instruction tuning on pre-trained vision-language models, while the LLaVA series (Liu et al., 2023b;a; Li et al., 2024a) advanced perception and reasoning via large-scale visual instruction tuning. Several recent studies (Doveh et al., 2024; Gavrikov et al., 2024; Lin et al., 2024; Huang et al., 2024; Mirza et al., 2025; Hansen et al., 2025; Mei et al., 2025) have advanced these models by introducing improved training and adaptation strategies. Recent models extend these capabilities: Qwen-2.5 VL (Bai et al., 2025) offers precise object localization, dynamic resolution, and agentic tool execution; InternVL3 (Zhu et al., 2025) improves perception and reasoning through domain-specific pretraining on 3D scenes, GUIs, and video; Gemma 3 (Team et al., 2025), Ovis 2 (Lu et al., 2025), and GLM 4.5V (Hong et al., 2025) demonstrate strong performance across diverse multimodal benchmarks. High-resolution variants such as InternLM XComposer 2.5 (Zhang et al., 2024a) and VILA HD 4K (Shi et al., 2025), enable detailed perception and document processing. Proprietary models like GPT-5 (OpenAI, 2025) and Gemini 2.5 Pro (Comanici et al., 2025) set the state-of-the-art on complex multimodal tasks through large-scale training and enhanced reasoning.

These LMMs form the foundation for evaluating multimodal reasoning over scientific documents. In PRISMM-Bench, we benchmark 21 top-performing models to detect, understand, and correct real-world inconsistencies in peer-reviewed papers, exposing both their strengths and limitations.

**Multimodal Benchmark on Scientific Paper Understanding.** Prior benchmarks often focus on isolated scientific elements such as diagrams (Kafle et al., 2018; Chaudhry et al., 2020; Kahou et al., 2018), charts (Masry et al., 2022; Methani et al., 2020), or tables (Cheng et al., 2022; Nan et al., 2022). Recent datasets like MathVista (Lu et al., 2024), MathVerse (Zhang et al., 2024b), and ArXivQA (Li et al., 2024b) integrate multiple modalities, but still treat figures and equations in isolation rather than in full-paper context. Whole-paper QA resources such as PubMedQA (Jin et al., 2019), BioASQ (Krithara et al., 2023) and QASPER (Dasigi et al., 2021) provide human-written questions, yet these are mostly abstract-based and limited to yes/no or short-span answers.

Closer to our setting, QASA (Lee et al., 2023) provides 1.8K expert-written questions on ML papers, but remains text-only and does not require reasoning over figures or tables. SPIQA (Pramanick et al., 2024) introduces multimodal scientific QA, yet the questions are LLM-generated or human-annotated with an emphasis on information seeking, not grounded on expert reviews that often aim to critically evaluate scientific findings. SciDQA (Singh et al., 2024) is sourced from reviewer–author QA pairs, but it remains a text-only LLM benchmark without involving visual elements. In contrast, PRISMM-Bench is the first benchmark grounded in reviewer-flagged multimodal inconsistencies in scientific papers. Unlike prior work that isolates figures, tables, or text, our benchmark integrates visual and textual reasoning within the natural context of full research papers, while grounding tasks in authentic peer review feedback rather than synthetic or abstract-level annotations.

**Understanding of Inconsistencies.** Research on inconsistencies in language models spans prediction variance across paraphrased queries (Ravichander et al., 2020; Elazar et al., 2021) to factual inconsistency in summarization and long-form QA. To address the latter, prior work has introduced QA-based benchmarks (e.g., WikiContradict (Hou et al., 2024)), evaluation metrics (e.g., QAFactEval (Fabbri et al., 2022)), and detection methods based on QA (Wang et al., 2020), natural language inference (Lattimer et al., 2023), or probabilistic reasoning (Marinescu et al., 2025).

Closest to our setting, MMIR (Yan et al., 2025) evaluates multimodal models on artificially injected inconsistencies in materials such as slides and posters. In contrast, PRISMM-Bench introduces real-world reviewer-flagged inconsistencies in scientific papers. Rather than synthetic perturbations, our benchmark captures authentic challenges faced during scholarly review, spanning textual, visual, and cross-modal errors, and extends evaluation beyond detection to proposing remedies.

**Language Biases in Evaluation Benchmarks.** Multiple-choice evaluation is prone to linguistic biases, where models exploit surface-level patterns in answer options rather than reasoning over content. Prior studies show LLMs can achieve high accuracy even without the question, such as in TruthfulQA (Turner & Kurzeja, 2025), HellaSwag (Zellers et al., 2019), and ARC (Balepur et al., 2024). For example, Chandak et al. (2025) report 83% accuracy on TruthfulQA v2 using answer choices alone, with shortcut rates above 70% on HellaSwag. The recent trend of generating distractors with LLMs (e.g., in MMLU-Pro; Wang et al. (2024a)) can even exacerbate these artifacts.

To mitigate such biases, structured representations offer a promising direction. Analogous to authorship obfuscation in stylometry (Chinchor, 1998), structured formats remove stylistic and surface cues while retaining semantics. Drawing inspiration from FrameNet-based semantic parsing (Das et al., 2014) and MUC slot filling (Uchendu et al., 2023), PRISMM-Bench introduces JSON-based answer representations that encode key elements for capturing inconsistencies in scientific papers. This design reduces artifacts in answer choices and compels models to engage with multimodal content rather than exploiting linguistic shortcuts.

## 3    PRISMM-BENCH

PRISMM-Bench is built through a six-stage pipeline (Fig. 2): (1) review sourcing, (2) LLM-based review filtering, (3) manual annotation of reviewer-flagged inconsistencies (Sec. 3.1), (4) LMM-based task generation, (5) manual verification to finalize benchmark tasks (Sec. 3.2), and (6) LLM-based debiasing to reduce language biases (Sec. 3.3). The evaluation step is introduced in Sec. 3.4.

### 3.1    COLLECTION OF REVIEWER-FLAGGED INCONSISTENCIES

To build a benchmark of realistic and authentic inconsistencies, we sourced cases flagged by reviewers on OpenReview (ope, b), where comments often highlight discrepancies between textual content and visual or mathematical components, including figures, tables, and equations.

**Review Sourcing Strategy.** We collected reviews from ICLR 2024 & 2025 submissions via the OpenReview API v2 (ope, c). To maximize the likelihood that flagged inconsistencies persisted in the final public PDFs, we restricted to rejected or withdrawn submissions without rebuttals.[1] This yielded 18,009 reviews (details in App. D.1).

**LLM Review Filtering.** As manual screening for all reviews was infeasible, we employed an LLM for review filtering. Specifically, we used *Mistral Nemo 2407* (Mistral, 2024) with low temperature settings to summarize reviews and identify potential inconsistency mentions, resulting in a curated set of 6,056 potential inconsistencies spanning 2,458 reviews (prompt details in App. F.1).

**Manual Verification.** We performed a manual annotation pass using a custom web-based annotation tool. The tool presented the annotator with one reviewer-flagged inconsistency at a time, alongside the corresponding paper in an embedded PDF viewer. Annotators (1) verified whether reviewer comment described a factual and identifiable inconsistency, and (2) annotated the relevant textual and/or visual parts of the paper. For visual elements, the annotator selected and cropped regions from the PDF. For textual elements, they specified the page, line, and content. In addition, each inconsistency was assigned a category and a brief description in the annotator's own words. The tool logged annotations together with the original reviewer's comment and automatically collected metadata such as the crop bounding boxes in a structured format. Full details of the annotator background, annotation tool, captured metadata, annotation criteria and schema are provided in App. G.

---

[1]Our earlier exploration of review sourcing revealed that most reviewer-flagged inconsistencies were resolved during rebuttal and did not persist in the final versions, motivating the current refined sourcing strategy.

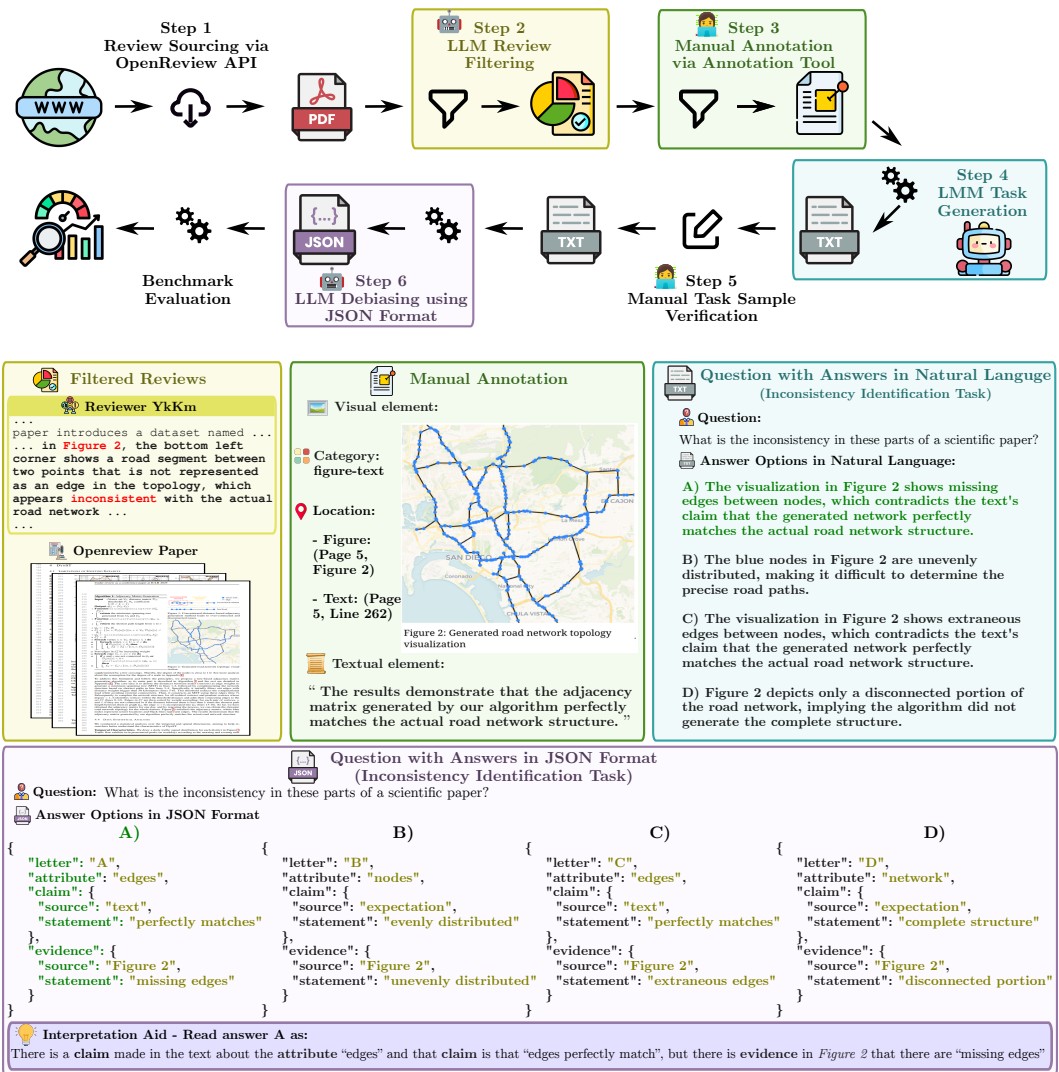

Figure 2: **Pipeline of PRISMM-Bench.** The top row illustrates the six main steps: (1) review sourcing, (2) LLM-based review filtering, and (3) manual annotation of metadata for reviewer-flagged inconsistencies (Sec. 3.1), (4) LMM-based task generation, (5) manual verification to construct benchmark tasks (Sec. 3.2), and (6) LLM-based debiasing to mitigate language biases (Sec. 3.3). The bottom row shows representative outputs at each stage: filtered reviews after step 2, inconsistency annotation after step 3, an example multiple-choice question in natural language after step 4, and its debiased JSON-format counterpart after step 6.

This process produced 384 validated inconsistencies across 353 ICLR submissions. We identified 15 categories of inconsistencies based on the elements involved (distribution shown in Fig. 9). The most common cases were intra-figure inconsistencies (23.7%) and figure-text mismatches (21.9%).

## 3.2 GENERATION OF BENCHMARK TASKS

From the verified inconsistencies, we constructed three multiple-choice tasks with four options (one correct, three distractors), following the evaluation choice of most recent frontier model releases (Yang et al., 2025a; Liu et al., 2025; Comanici et al., 2025; Team et al., 2024) and benchmarking efforts (Wang et al., 2024b; Zhang et al., 2025; Shabtay et al., 2025). We design the following three multiple-choice tasks.

**Inconsistency Identification (*Ident*).** The first task evaluates a model's ability to recognize inconsistencies within the given paper context, framed by the question: *"What is the inconsistency in"*

*these parts of a scientific paper?"* We adopt this generic question style because our preliminary study showed that sample-specific questions (e.g. *"What inconsistency is observed between Figure 2 and the accompanying text regarding the generated road network?"*) reveal the inconsistency content and oversimplify the task (see App. D.4.2 for details).

Candidate answers were generated using *Gemini 2.5 Flash* based on inconsistency descriptions and corresponding multimodal context. The answers were manually refined to ensure (1) the correct choice captured the inconsistency precisely and (2) the distractors are contextually relevant and plausible, but incorrect. We show an example of the *Ident* task in Fig. 2.

**Inconsistency Remedy (*Remedy*).** This task extends beyond simple detection by requiring models to how to fix the inconsistency by asking the question *"What action needs to be taken to resolve the inconsistency in these parts of a scientific paper?" Gemini 2.5 Flash* was employed to reformulate the inconsistency statements from *Ident* into specific, actionable remedy formulations. This task evaluates whether models can propose plausible solutions, rather than merely spotting inconsistencies.

**Inconsistency Pair Match (*Match*).** This task is built on a subset of inconsistencies that involve two distinct visual elements within a paper (192 samples). Given one element as context, the model must select its inconsistency counterpart from four options. By restricting the task to visual-visual mismatches, we specifically assess a model's ability to detect representation errors without relying on textual cues, simulating the common peer-review challenge of cross-checking figures and tables for consistency.

More details about the task generation process and a validation of the multiple-choice format against open-ended evaluation are provided in App. D.4 and App. C.3, respectively. We provide qualitative examples of the three tasks in App. A and a dataset viewing tool in the supplementary materials.

## 3.3 ALLEVIATION OF LINGUISTIC BIASES

During pilot experiments, we observed that models achieved well above random accuracy even when the visual context was withheld. For example, *Gemini 2.5 Flash* reached 57.6% accuracy on the *Ident* task without context (vs. 25% random chance). This indicated that models exploit linguistic priors and surface patterns in the answer options rather than reasoning over the actual content. Further analysis showed that factors such as answer length, relative position, and phrasing contributed to this bias, echoing known challenges in multiple-choice design (Gierl et al., 2017).

To combat this bias, we first tried refining the distractors with text manipulation, which proved insufficient. Therefore, we introduced structured representations that minimize natural language cues. We designed the *Evidence–Claim JSON* format for the *Ident* task and the *Target–Action JSON* format for the *Remedy* task. Converting answers into these structured formats using an LLM reduced *Gemini 2.5 Flash's* no-context accuracy on the *Ident* task to 34.0%. We manually verified a 20% subset of the inconsistencies to ensure the semantic fidelity of the JSON-formatted answers. An example of the Evidence-Claim JSON format for the *Ident* task is provided in Fig. 2. Full details on our debiasing procedure and the structured formats are provided in App. D.4.2.

This design choice is further supported by our user study (Sec. 4.4), which reveals that humans rely minimally on linguistic priors. In contrast, models evaluated on natural language options maintain high accuracy without context, exposing a fundamental mismatch in evaluation fidelity. By adopting structured JSON representations, we align model evaluation conditions more closely with human cognitive constraints, suppressing surface-level shortcuts and enabling a fairer assessment of true multimodal reasoning.

## 3.4 CONTEXTUAL GRANULARITY IN EVALUATION

We evaluate model performance under three levels of contextual granularity, reflecting different real-world reading conditions and reasoning demands.

**Focused Context (*Focused*).** The model is presented only with the minimal necessary components — an extracted visual element (e.g., cropped figure or table) and/or the precise text passage (e.g.,

sentence or paragraph) involved in the inconsistency, as annotated. This setting isolates the key content, testing the model's ability to detect inconsistencies with minimal noise.

**Page Context (*Page*).** The model receives a 144 DPI rasterized image of the entire page(s) where the inconsistency occurs. Visual elements are not pre-cropped, requiring the model to locate and interpret relevant content within the full page layout. This simulates realistic reading conditions where inconsistencies must be identified without prior localization.

**Document Context (*Document*).** The model receives the entire scientific paper as a sequence of page images. To accommodate architectural constraints, we follow MMLongBench-Doc (Ma et al., 2024) and segment the document into collages: a total of 5 images are fed to the model, each containing $n_{pages}/5$ pages arranged in a 3-column grid. This setting evaluates the model's capacity for long-range, cross-page reasoning and document-level grounding. For models with high-resolution processing constraints, such as *LLaVA Onevision (7B, 72B)*, we reduce input to 3 images with $n_{pages}/3$ pages each to avoid exceeding the context window.

# 4 EXPERIMENTS

## 4.1 EXPERIMENTAL SETUP

**Model Selection.** We evaluate 21 LMMs spanning a diverse range of architectures: 16 open-weight models of varying scales, two specialized high-resolution models, and three proprietary models. Selection was guided by performance on the Open VLM Leaderboard (ope, a) and the availability of the latest model versions.

**Inference Details.** To ensure consistent scoring, we enforced a strict answer format. Prompts at both the system and user level instructed models to output only the letter corresponding to their chosen option. For reasoning-enabled models, answers were extracted separately from reasoning traces, enclosed in `<think></think>` tags, before postprocessing against the ground truth.

All open-weight models were grouped into three parameter count categories and evaluated with `vLLM` v0.10.1 (vll). Experiments were conducted on 4×NVIDIA A100 64GB GPUs with greedy decoding, except for InternVL3.5 (8B, 38B) which required a temperature of 0.6 for stable reasoning. Proprietary models were accessed via their official APIs with greedy decoding except *GPT-5 (minimal, high)* which has a fixed temperature of 1.0.

Each model was evaluated on all three tasks (*Ident*, *Remedy*, *Match*), and across three contextual granularity levels (Sec. 3.4). For *Match*, only *Focused Context* was used. This design yields seven evaluation configurations per model, enabling fine-grained analysis of how model architecture, scale and input context affect inconsistency detection performance.

## 4.2 MAIN RESULTS

Table 1 summarizes the aggregate performance of all evaluated models. Our results reveal clear trends in how LMMs handle inconsistency detection and correction.

**Performance Landscape.** Proprietary models substantially outperform their open-weight counterparts. GPT-5 (high reasoning) reached the highest average performance of 53.9%. By contrast, the best open-weight model GLM 4.5V 106B achieves 42.5%, matching GPT-5 (minimal reasoning) but trailing its high-reasoning variant by 11.4 points. These results underscore the difficulty of the benchmark: even the best-performing models remain far from the reliability required of automated scientific assistants.

**Impact of Context and Task Formulation.** Performance drops consistently as context scope expands. Models achieve their best accuracy in the *Focused* setting but often degrade toward random chance under *Page* and *Document* inputs, reflecting persistent challenges with distraction and long-range grounding in dense, multi-page inputs. To rule out input quality effects, we performed an ablation study on rasterization resolution, confirming our 144 DPI baseline (cf. App. C.1).

Task formulation also plays a critical role. *Remedy* scores are consistently lower than *Ident*, showing that proposing corrections requires deeper reasoning than only detection. Performance on *Match*

Table 1: Accuracy (%) of 21 diverse LMMs across three tasks (*Ident*, *Remedy*, *Match*) and three levels of contextual granularity (Sec. 3.4). For *Match*, results are reported only under the *Focused* setting. Best result per task bolded, second best underlined. [R] denotes reasoning models.

| Model | Params [B] | Focused Ident (384) | Focused Remedy (384) | Focused Match (192) | Page Ident (384) | Page Remedy (384) | Document Ident (384) | Document Remedy (384) | Average (960) |
|---|---|---|---|---|---|---|---|---|---|
| *Small Open-Weight Models (<9B)* | | | | | | | | | |
| Gemma 3 4B | 4.0 | 27.9 | 29.9 | 39.6 | 25.0 | 24.7 | 26.6 | 27.1 | 27.8 |
| LLaVA OV 7B | 7.0 | 30.5 | 28.4 | 29.7 | 32.0 | 28.4 | 28.1 | 27.9 | 29.2 |
| Ovis2 8B | 8.0 | 35.4 | 29.4 | 22.4 | 34.4 | 27.3 | 31.8 | 28.1 | 30.4 |
| Qwen 2.5 VL 7B | 7.0 | 32.8 | 31.3 | 58.9 | 29.9 | 29.7 | 26.8 | 27.1 | 31.9 |
| InternVL3 8B | 8.0 | 36.5 | 29.4 | 56.3 | 28.6 | 27.6 | 30.7 | 31.8 | 32.7 |
| InternVL3.5 8B[R] | 8.0 | 49.5 | 35.9 | 45.8 | 38.3 | 30.5 | 36.7 | 31.0 | 37.7 |
| *Medium Open-Weight Models (9B—38B)* | | | | | | | | | |
| Gemma 3 27B | 27.0 | 36.2 | 32.8 | 59.9 | 30.7 | 28.6 | 31.0 | 27.3 | 33.3 |
| Gemma 3 12B | 12.0 | 33.9 | 30.5 | 63.5 | 30.7 | 25.8 | 30.7 | 30.5 | 32.9 |
| Qwen 2.5 VL 32B | 32.0 | 42.4 | 37.0 | 45.8 | 37.2 | 34.6 | 38.3 | 27.9 | 37.0 |
| Ovis2 34B | 34.0 | 50.0 | 41.1 | 37.0 | 40.6 | 36.2 | 33.3 | 31.8 | 38.7 |
| InternVL3 38B | 38.0 | 46.6 | 38.5 | 56.8 | 40.6 | 35.7 | 37.0 | 32.0 | 39.8 |
| InternVL3.5 38B[R] | 38.0 | 54.4 | 43.5 | 50.9 | 40.9 | 31.3 | 33.9 | 31.5 | 39.8 |
| *Large Open-Weight Models (>38B)* | | | | | | | | | |
| LLaVA OV 72B | 72.0 | 35.4 | 30.5 | 28.1 | 32.3 | 28.4 | 31.5 | 26.0 | 30.5 |
| Qwen 2.5 VL 72B | 72.0 | 49.7 | 37.2 | 32.8 | 44.0 | 33.3 | 35.4 | 25.3 | 37.1 |
| InternVL3 78B | 78.0 | 49.5 | 39.3 | 45.3 | 39.3 | 33.9 | 35.9 | 30.5 | 38.6 |
| GLM 4.5V 106B[R] | 106.0 | 51.8 | 43.2 | 52.1 | 45.8 | 35.9 | 40.9 | 33.1 | 42.6 |
| *Specialized High-Resolution Models* | | | | | | | | | |
| InternLM XC 2.5 7B | 7.0 | 28.4 | 25.3 | 27.6 | 29.9 | 27.1 | 29.9 | 28.6 | 28.2 |
| VILA HD 4K 8B | 8.0 | 31.0 | 30.7 | 25.5 | 30.2 | 29.2 | 28.6 | 28.4 | 29.4 |
| *Proprietary Models* | | | | | | | | | |
| GPT-5 (minimal)[R] | — | 53.6 | 43.5 | 63.0 | 47.1 | 36.5 | 40.9 | 32.8 | 44.0 |
| Gemini 2.5 Pro[R] | — | 65.9 | 61.2 | 66.7 | 54.7 | 51.8 | 39.8 | 36.7 | 52.8 |
| GPT-5 (high)[R] | — | 63.8 | 54.4 | 70.3 | 58.1 | 51.0 | 46.9 | 41.1 | 53.9 |

varies widely across models: Gemma 3 12B achieves 63.5%, rivaling proprietary models, whereas much larger models such as InternVL3 78B lag behind at 45.3%. These results suggest that architectural design, not just scale, is critical for relational reasoning.

**Model Characteristics.** Reasoning-enabled models show benefit. For example, InternVL3.5 8B outperforms its non-reasoning predecessor InternVL3 8B by over 5 percentage points, achieving accuracy comparable to models with nearly nine times more parameters. Turning off chain-of-thought reduces accuracy by up to 14 points (cf. App. C.2), directly confirming the contribution of reasoning. In contrast, high-resolution specialists (VILA HD 4K 8B, InternLM XC 2.5 7B) show little advantage in extended-context settings. More broadly, our results challenge the "bigger is better" paradigm: scaling up parameter counts alone does not guarantee higher performance, with diminishing returns observed from medium- to large-scale models.

Overall, these findings highlight the current limitations of LMMs for scientific document analysis. Future progress will require advances in reasoning architectures to move beyond error detection toward correction, as well as more robust mechanisms for grounding over long, distractive contexts.

## 4.3 ANALYSIS OF CHAIN-OF-THOUGHT REASONING

Reasoning variants perform better than non-reasoning counterparts and reach results comparable to much larger models. For instance, InternVL3.5 8B achieves an average of 37.7%, rivaling large open-weight models and surpassing several 72B non-reasoning models. We therefore study how reasoning-enabled models leverage chain-of-thought (CoT) to improve performance on PRISMM-

Bench. To do so, we re-evaluated a selection of reasoning models on the *Ident* task with *Focused Context* with reasoning turned off and compared the performance.

**Ablation Results.** Disabling reasoning reveals the critical role of CoT in detecting subtle inconsistencies. For example, GLM 4.5V drops from 51.8% to 43.2% (-16.6%), InternVL3.5 8B from 49.5% to 40.6% (-18.0%), and InternVL3.5 38B suffers the largest decline, from 54.4% to 40.4% (-25.7%).

**Why Reasoning Helps.** To understand these performance differences, we focused on inconsistencies where InternVL3.5 38B succeeded with reasoning but failed without. We identified three consistent patterns: (1) Structured input handling: Reasoning-enabled models interpret the JSON-formatted options in natural language, clarifying subtle distinctions without exploring linguistic biases (cf. low without-context performance for reasoning models in Sec. 4.4). (2) Cross-modal grounding: CoT traces show models explicitly reasoning over both text and visuals, breaking complex information into smaller units and reusing them later in the reasoning chain. (3) Concept linking: Reasoning enables models to connect fine-grained context with domain knowledge and abstract concepts, allowing stronger logical inference beyond surface pattern recognition. We provide a detailed case study illustrating these effects in App. C.2.

## 4.4 User Study and Linguistic Bias Analysis

To complement our benchmark, we conducted a user study to establish a human baseline and quantify *visual reliance* — the extent to which answers depend on genuine multimodal reasoning rather than linguistic shortcuts. While our benchmark uses structured JSON outputs for models, our participants are evaluated on natural language (NL) questions as structured formats are less practical without prior training. To enable direct comparison, we re-evaluated representative LMMs on the same *Ident* task subset using natural language answer options.

**Setup.** Eight non-author participants with at least PhD-level computer science research experience each answered ten randomly sampled *Ident* task questions: five in *Focused* context and five in *Document* context. We provide a detailed analysis of the representativeness of this subset in App. E.2. For each question, participants first answered without context (question + answer options only), then with context. *Focused* context consisted of cropped images and/or text excerpts; *Document* context contains links to original PDFs. The survey was implemented via a custom web interface (cf. App. E.1).

Table 2: User Study Results. For each context scope, we report Accuracy (%) for both Natural Language (NL) and JSON answer options. Human performance with NL is shown for reference.

| Model | Without Context | | Focused Context | | Whole Document Context | |
|---|---|---|---|---|---|---|
| | NL | JSON | NL | JSON | NL | JSON |
| *Human* | 27.5 | — | 77.5 | — | 65.0 | — |
| InternVL3.5 8B[R] | 49.3 | 28.4 | 76.3 | 47.4 | 56.8 | 35.1 |
| InternVL3.5 38B[R] | 53.7 | 25.3 | 76.3 | 71.1 | 70.3 | 40.5 |
| Qwen 2.5 VL 72B | 47.8 | 38.8 | 65.8 | 65.8 | 43.2 | 48.6 |
| Gemini 2.5 Pro[R] | 70.1 | 37.3 | 81.6 | 65.8 | 83.8 | 37.8 |

**Analysis.** As shown in Table 2, top models like Gemini 2.5 Pro exceed humans under *Focused Context* (81.6% vs. 77.5%) and *Whole Document Context* (83.8% vs. 65.0%). However, a crucial difference emerges in the *Without Context* condition: LMMs maintain high accuracy (up to 70.1%), whereas human performance drops near chance (27.5%). This indicates that LMMs rely heavily on linguistic regularities that humans cannot exploit. Switching to JSON formatting neutralizes this advantage. Without context, model performance collapses toward human levels (e.g., InternVL3.5 38B drops from 53.7% to 25.3%). With context and JSON-structured answer, LMMs no longer match human NL performance, confirming that linguistic shortcuts inflate perceived model capability.

Table 3: Impact of answer representation on without-context performance and visual reliance. Accuracy is reported for the *Ident* task. $R$ is computed according to Eq. 1. using *Focused* context as the with-context baseline.

| Model | Natural Language | | JSON | |
|---|---|---|---|---|
| | **Accuracy** | $R$ | **Accuracy** | $R$ |
| InternVL3.5 8B | 45.6 | 17.1 | 28.6 | 29.3 |
| InternVL3.5 38B | 52.9 | 22.5 | 26.3 | 38.1 |
| Qwen 2.5 VL 72B | 49.7 | 16.1 | 36.5 | 20.8 |
| Gemini 2.5 Pro | 61.2 | 43.8 | 37.8 | 45.2 |

To quantify how much models and human rely on visual evidence versus linguistic priors, we compute the **Visual Reliance Ratio** $R$, adapted from the normalized Perceptual Score (Gat et al., 2021):

$$R = \frac{Acc_{\text{with\_context}} - Acc_{\text{without\_context}}}{1 - Acc_{\text{without\_context}}} \tag{1}$$

Higher $R$ indicates stronger dependence on visual context. Human participants achieve $R = 69.0\%$, while the top model (InternVL3.5 8B) achieves $R = 53.5\%$, confirming that humans rely more on visual grounding than current LMMs.

**Probing Linguistic Bias.** To confirm this effect generalizes beyond the user study subset, we re-evaluated the same four representative LMMs on the full *Ident* dataset under both Natural Language and JSON formats (Table 3). The same pattern holds: under natural language, models achieve inflated accuracies without context (e.g. 61.2% for Gemini 2.5 Pro) but performance drops toward chance under JSON. Correspondingly, $R$ increases under JSON for all models, showing that structured outputs suppress linguistic shortcuts and force models to rely more on visual evidence.

**Insights.** Two key conclusions emerge: (1) MCQs with long-form answers in natural language overstate LMM performance, as models can exploit linguistic regularities imperceptible or irrelevant to humans. (2) Structured JSON representations mitigate this bias, revealing that even the strongest LMMs still fall short of human-level visual grounding and rely on surface cues when available.

## 5 CONCLUSIONS

We introduce PRISMM-Bench, a multimodal benchmark for evaluating LMMs on real-world scientific inconsistencies. We show that even top-performing models struggle with cross-modal reasoning and long-context grounding, while structured answer formats mitigate linguistic shortcuts. This work highlights limitations of LMMS as scientific assistants and motivates future improvements in filtering pipelines, cross-domain datasets, and debiasing strategies for long-form MCQs evaluation.

**Limitations.** Our benchmark is limited in scope: it currently focuses on AI-domain papers from ICLR 2024 & 2025 and emphasizes rejected submissions to capture unresolved errors. As a result, both the scale and domain coverage are restricted. Future work should expand to other fields and venues, and explore inconsistencies that may persist in accepted papers, offering a broader and more representative testbed.

## 6 ETHICS STATEMENT

This work introduces PRISMM-Bench, a benchmark for evaluating multimodal large language models (MLLMs) on scientific document understanding. In developing the benchmark, we exclusively use publicly available research papers from ICLR 2024 & 2025, which are distributed under the Creative Commons Attribution 4.0 (CC-BY 4.0) license. This license explicitly permits redistribution, remixing, and adaptation of the material with proper attribution, and we ensure that all source materials are used in full compliance with these terms.

Our study also includes a small-scale user study to establish a human baseline. All participants were experienced researchers, voluntarily consented to take part, and no personally identifying information was collected or reported. The study design posed no foreseeable risks to participants and did not involve vulnerable populations.

We recognize that benchmarks can influence the direction of future model development. While PRISMM-Bench may highlight weaknesses in existing systems, it is not intended to facilitate misuse, such as adversarial attacks on models, but rather to promote more robust and trustworthy scientific document analysis. We release the benchmark with the goal of supporting transparent, reproducible, and ethical research, in line with the ICLR Code of Ethics.

No conflicts of interest, sensitive data, or privacy concerns arise in this work.

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

APPENDIX

In the appendix, we first discuss illustrative **qualitative examples** (App. A), including examples of text-table and figure-equation inconsistencies. We then provide a comprehensive **list of assets** (App. B), detailing the data sources, licenses, and models used, including both open-source and proprietary ones. The **ablations section** (App. C) explores the impact of rasterization resolution on model performance, extends the analysis of the effectiveness of Chain-of-Thought (CoT) reasoning with a detailed case study, and validates our choice of the multiple-choice question format against open-ended evaluation. The **dataset construction section** (App. D.1) explains our refined methodology for sourcing, filtering, and annotating inconsistencies from scientific papers, including a discussion of the annotation criteria, a custom-built annotation tool, and dataset statistics. Next, we detail the **process of generating LLM-based questions** (App. D.4) for our benchmark tasks (Inconsistency Identification, Remedy, and Pair-Match), including our debiasing strategies using structured JSON representations. Finally, we provide details on the **user study implementation and representativeness** (App. E.1), the **full LLM prompts** (App. F) used for various tasks, and **screenshots** (App. G and H) of our annotation and survey applications to provide a clear understanding of our methodology.

## A    QUALITATIVE EXAMPLES

We show qualitative examples of a text–table inconsistency in Fig. 3 and a figure-equation inconsistency in Fig. 4, together with their corresponding evaluation tasks.

## B    LIST OF ASSETS

Our images and annotations are sourced from publicly available datasets, and we distribute our data in compliance with the licensing terms of the original sources.

The document and review data source can be found here:

- ICLR 2024 on OpenReview (https://openreview.net/group?id=ICLR.cc/2024/Conference): All papers were released under the CC BY 4.0 license.
- ICLR 2025 on OpenReview (https://openreview.net/group?id=ICLR.cc/2025/Conference): All papers were released under the CC BY 4.0 license.

The list of source code and model weights can be found here:

- Qwen2.5-VL (https://github.com/QwenLM/Qwen2.5-VL): Released under the Apache-2.0 license.
- LLaVA-NeXT (https://github.com/LLaVA-VL/LLaVA-NeXT): Released under the Apache-2.0 license.
- Gemma 3 (https://github.com/google-deepmind/gemma): Released under the Apache-2.0 license.
- Ovis 2 (https://github.com/AIDC-AI/Ovis): Released under the Apache-2.0 license.
- InternVL (https://github.com/OpenGVLab/InternVL): Released under the MIT license.
- GLM-V (https://github.com/zai-org/GLM-V): Released under the Apache-2.0 license.
- Mistral NeMo (https://github.com/mistralai/mistral-inference: Released under Apache-2.0 license
- vLLM (https://github.com/vllm-project/vllm): Released under the Apache-2.0 license.
- MinerU (https://github.com/opendatalab/MinerU): Released under the AGPL-3.0 license.

The list of proprietary models used can be found here:

- Google Gemini (https://deepmind.google/models/gemini/flash/): Used in version Gemini Flash 2.5 and Gemini Pro 2.5, released on June 17, 2025.

- OpenAI GPT (https://github.com/LLaVA-VL/LLaVA-NeXT): Used in version GPT-5, released on August 7, 2025.

## C    ABLATIONS

### C.1    IMPACT OF RASTERIZATION RESOLUTION

Scientific papers contain dense text and fine-grained visual elements such as axis labels, annotations, and subscripts, which are often crucial for detecting subtle inconsistencies. To test whether rasterization resolution impacts detection performance, we varied the DPI used to extract images from the PDF and evaluated a representative set of strongest proprietary and open-weight models of different sizes on the *Inconsistency Ident* task with *Focused Context*, keeping all other settings fixed.

Table 4: Accuracy of LMMs under different rasterization resolutions. Results are reported for the *Ident* task with *Focused Context*. Percentage change is calculated relative to the 144 DPI baseline.

| DPI | VILA HD 4K 8B | | InternVL3.5 8B | | InternVL3.5 38B | | Ovis2 34B | | Gemini 2.5 Pro | |
|-----|---------|----------|---------|----------|---------|----------|---------|----------|---------|----------|
|     | Default | % Change | Default | % Change | Default | % Change | Default | % Change | Default | % Change |
| 72  | 31.3 | +1.3 | 42.7 | -12.7 | 45.8 | -20.1 | 43.1 | -13.1 | 67.2 | -3.3 |
| 144 | 30.9 | –    | 48.9 | –     | 57.3 | –     | 49.6 | –     | 69.5 | –    |
| 300 | 29.4 | -4.9 | 45.4 | -7.2  | 51.9 | -9.4  | 50.4 | +1.6  | 70.6 | +1.6 |

**Low resolutions harm performance.** Most models showed significant drops at 72 DPI, up to -20.1% for InternVL3.5 38B. Open-weight models were generally more vulnerable, though even Gemini 2.5 Pro declined by -3.3%. Surprisingly, VILA HD, despite being trained for high-resolution inputs, showed a slight accuracy gain at this lower setting.

**Higher resolutions do not always improve accuracy.** Increasing from 144 to 300 DPI yielded mixed outcomes. While Gemini 2.5 Pro and Ovis2 34B benefited slightly, InternVL models performed worse, and VILA HD again failed to leverage the higher fidelity despite its specialized training. This suggests that additional detail can sometimes overwhelm global reasoning or misalign with training distributions.

**Resolution sensitivity is architecture-specific.** Overall, the assumption that higher resolution improves inconsistency detection does not hold universally. Performance varies with model design and pretraining data, and high-resolution training does not guarantee an edge in handling scientific inconsistencies. Careful DPI control is critical for fair evaluation. In our benchmark, 144 DPI provides a practical balance of visual clarity, computational cost, and cross-model comparability.

### C.2    CASE STUDY: CHAIN-OF-THOUGHT REASONING

To illustrate the effect of chain-of-thought reasoning discussed in Section 4.3, we present a representative example in Fig. 5. The figure reports results for *Unique Successful Jailbreaks*, a count-based metric that is strictly non-negative. However, some error bars extend below the zero line on the y-axis—an inconsistency that invalidates the figure.

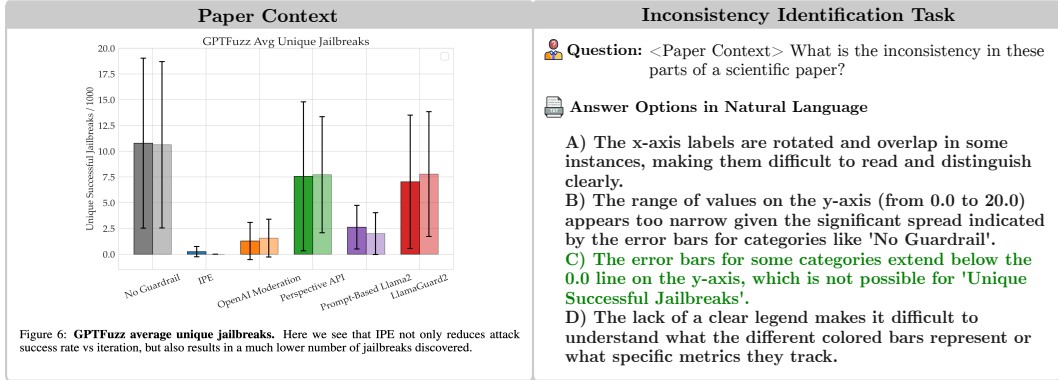

Figure 5: Inconsistency example for case study. Right: Visual context. Left: Question and answer options for *Ident* task. Natural language options used for ease-of-comprehension, LMM was tasked using JSON.

Without reasoning, InternVL3.5 38B selected the distractor "x-axis labels overlap," (option A) justifying it with a generic but factually incorrect critique, as the labels were perfectly legible. The model defaulted to a template-like response rather than verifying claims against visual evidence.

In contrast, the reasoning-enabled model produced a systematic chain-of-thought: (1) ruling out label overlap (by observing the labels were *'spaced out and readable'*), (2) confirming the y-axis range was sufficient (noting all data was *'within the 0-20 range'*), (3) dismissing legend critiques (since a legend *'isn't necessary'* when bars are directly labeled), and (4) crucially, identifying the logical error of error bars that *'shouldn't go below zero if the metric [...] can't be negative'*). This stepwise elimination and domain-aware inference led to the correct answer. The full reasoning chain of InternVL3.5 38B is available in Fig. 6.

This case highlights two key strengths of reasoning: (1) systematic elimination of distractors, and (2) integration of domain knowledge (e.g. non-negativity of counts) with visual grounding. Although reasoning increase output length (average of 473 tokens per query in our case), it substantially improves multimodal consistency and robustness, making CoT a key mechanism for handling subtle scientific document inconsistencies.

## C.3 VALIDITY OF MULTIPLE-CHOICE VS. OPEN-ENDED EVALUATION

While open-ended generation is closer to real-world usage, the choice of a multiple-choice question (MCQ) format for our benchmark was motivated by the need for a reliable, reproducible, and bias-controlled evaluation protocol. To validate this choice, we conducted additional experiments comparing our MCQ format with an open-ended evaluation using an LLM-as-a-judge approach.

**Setup.** We tested a representative subset of the identification task using open-ended responses from eight models. Their outputs were rated by three different judges—Gemini 2.5 Pro, GPT-5 (high), and GLM 4.5V 106B—on a Likert-5 scale measuring semantic alignment with the ground-truth inconsistency. Each judge was evaluated both with and without *Focused Context*.

**Analysis of Open-Ended Evaluation.** As shown in Table 5, our open-ended evaluation revealed several critical limitations that undermine its suitability for a standardized benchmark. First, we observed significant **inter-judge inconsistency**, where LLM judges produced diverging scores and rankings for identical model outputs. For instance, Gemini 2.5 Pro and GPT-5 frequently disagreed on absolute scores, leading to rank shifts of two to three positions for certain models depending on the judge employed. Furthermore, GLM 4.5V 106B proved particularly unstable in its role as a judge, further complicating reproducibility.

A second challenge relates to **non-deterministic scoring** in proprietary reasoning models. Since a temperature of zero is not always available for models such as GPT-5, repeated evaluations of the same answer do not yield deterministic results. This introduces noise into the benchmarking process, which is especially problematic when performance differences between high-performing

Table 5: LLM-as-a-judge evaluation. Scores are average Likert ratings (1-5); percentages indicate scores > 3. R denotes the model's rank per judge.

| Candidate Model | Gemini 2.5 Pro | | | | GPT-5 | | | | GLM 4.5V 106B | | | |
| --- | --- | --- | --- | --- | --- | --- | --- | --- | --- | --- | --- | --- |
| | w/o context | | context | | w/o context | | context | | w/o context | | context | |
| | Score | R | Score | R | Score | R | Score | R | Score | R | Score | R |
| Gemini Pro 2.5[R] | 2.80 (44.0%) | 2 | 3.06 (54.0%) | 1 | 2.71 (40.0%) | 2 | 2.80 (44.0%) | 2 | 3.06 (54.0%) | 1 | 3.50 (62.0%) | 1 |
| GPT-5 (high)[R] | 3.00 (54.0%) | 1 | 2.70 (42.0%) | 2 | 2.76 (52.5%) | 1 | 2.94 (52.0%) | 1 | 3.02 (51.0%) | 2 | 3.32 (58.0%) | 2 |
| InternVL 3.5 38B[R] | 2.22 (34.0%) | 6 | 2.26 (34.0%) | 4 | 2.06 (25.0%) | 4 | 1.98 (24.4%) | 4 | 2.68 (48.0%) | 3 | 2.84 (46.0%) | 3 |
| Qwen 2.5 VL 72B | 2.46 (38.0%) | 3 | 2.40 (38.0%) | 3 | 2.30 (28.3%) | 3 | 2.13 (26.7%) | 3 | 2.42 (34.0%) | 6 | 2.60 (40.0%) | 6 |
| GLM 4.5V 106B[R] | 2.40 (40.0%) | 4 | 2.08 (28.0%) | 5 | 2.04 (22.0%) | 5 | 1.94 (22.0%) | 5 | 2.56 (42.0%) | 5 | 2.83 (47.9%) | 4 |
| InternVL 3.5 8B[R] | 2.04 (28.0%) | 7 | 1.94 (28.0%) | 6 | 1.94 (18.0%) | 7 | 1.66 (18.0%) | 8 | 2.60 (42.0%) | 4 | 2.78 (42.0%) | 5 |
| Gemma 3 12B | 2.24 (34.0%) | 5 | 1.78 (24.0%) | 8 | 2.03 (19.3%) | 6 | 1.78 (18.4%) | 6 | 2.22 (32.0%) | 7 | 2.50 (36.0%) | 7 |
| Ovis 2 34B | 1.96 (20.0%) | 8 | 1.86 (22.0%) | 7 | 1.83 (14.0%) | 8 | 1.74 (16.0%) | 7 | 2.18 (32.0%) | 8 | 2.30 (32.0%) | 8 |

models are small. Finally, we identified a clear **judge-model similarity bias**, where judges tended to rate outputs from their own model families more favorably. This phenomenon, documented in prior evaluations such as MT Bench (Zheng et al., 2023) and AlpacaEval (Dubois et al., 2025), makes it difficult to disentangle true model capability from stylistic alignment with the judge.

**Conclusion.** In contrast to open-ended evaluation, the MCQ format remains more reproducible, providing deterministic scoring, controlled difficulty through distractors, and identical evaluation conditions without dependence on third-party proprietary judges. Combined with our JSON-based debiasing strategies, the MCQ format provides a robust and reliable framework for PRISMM-Bench.

## D DATASET CONSTRUCTION

### D.1 REVIEW SOURCING

**Initial Exploration with Regex Matching.** Before finalizing the review sourcing strategy described in the main paper, we conducted an exploratory study to detect potential inconsistencies mentioned in reviews for ICLR 2024 using a simple regular expression (regex) approach. Reviews were accessed via the OpenReview API[2], focusing on the *"Weaknesses"* and *"Questions"* sections, which were most likely to contain critical feedback. Each sentence was parsed for co-occurrence of terms related to inconsistencies (e.g., "mismatch", "conflict") and references to visual elements (e.g., "figure", "table", "equation"). The pseudocode for this procedure is shown below:

---

**Algorithm 1** Pseudocode for regex matching

---

1: DEFINEPATTERN(inconsistency_pattern, `r'(inconsisten | mismatch | doesn['] t match | not match |conflict | discrepanc)'`)
2: DEFINEPATTERN(visual_pattern, `r'(figure | fig.? | table | graph | plot | image | diagram | equation)'`)

3: results ← [ ]
4: **for** each review in reviews **do**
5:     sections ← EXTRACTSECTIONS(review)
6:     **for** each section in sections **do**
7:         sentences ← SPLITINTOSENTENCES(section)
8:         **for** each sentence in sentences **do**
9:             **if** MATCHES(sentence, inconsistency_pattern) **and** MATCHES(sentence, visual_pattern) **then**
10:                 APPEND(results, sentence)
11:             **end if**
12:         **end for**
13:     **end for**
14: **end for**
15: **return** results

---

[2]https://docs.openreview.net/reference/api-v2

Manual inspection confirmed that reviews indeed contained valuable references to visual-textual mismatches. However, two limitations emerged: (1) regex captured only strict keyword formulations, missing paraphrased or indirect mentions of inconsistencies, and (2) many inconsistencies referenced papers that had been updated after rebuttal, making it impossible to locate the original errors in the PDF versions available through OpenReview.

**Refined Collection Strategy.** To address these issues, we refined our strategy in two ways:

1. **Conference selection:** We shifted our focus on papers without author rebuttals. This ensured that flagged inconsistencies were more likely to remain in the available PDFs.
2. **LLM filtering:** Instead of regex, we employed *Mistral Nemo 2407* at a low temperature to summarize reviews and extract candidate inconsistency statements. This approach captured non-strict formulations (e.g., "does not align with" instead of "mismatch") and produced structured outputs, making them easier to present to annotators in the verification interface.

This refinement reduced our initial pool of 120,329 ICLR reviews to 18,009 reviews. The LLM outputs were stored in structured JSON format, with each paper ID associated with a list of flagged inconsistencies.

**Example Output.** Throughout the appendix, we are going to illustrate our data preparation pipeline use the paper ID `vXSCD3ToCS`[3] as an example. We illustrate the example out after the LLM-assisted filtering in Fig. 7.

This structured representation provided a natural starting point for the subsequent manual verification stage described in App. D.2.

## D.2   ANNOTATION PROCESS

**Annotator Background.** The annotation was conducted by the first author, who has an advanced background in Computer Science and Machine Learning. A consistent annotation standard was maintained throughout the project; any ambiguous or borderline cases were discussed with senior researchers until a consensus was reached.

**Annotation Criteria.** During manual verification, the annotator judged each reviewer comment against the following criteria:

1. The comment reflects objective feedback rather than a subjective suggestion.
2. The comment describes an inconsistency involving two contradicting facts.
3. Both conflicting parts can be located in the PDF.
4. The inconsistency can be identified without deep domain-specific expertise (focus on visual/document-level inconsistencies).
5. The inconsistency is significant and factual, not a minor typo or stylistic choice.

**Annotation Interface.** We implemented a custom web-based tool in `Next.js`. The interface displayed the reviewer's comment (extracted by the LLM) alongside the corresponding paper embedded as a PDF viewer (compare Fig. 18, Fig. 19 for screenshots of the app's interface). The annotator could:

- Read the reviewer's comment and decide whether it fulfilled criteria (1) and (2). If not, the instance was skipped.
- Search and inspect the relevant region of the embedded PDF.
- Toggle between *one-part* and *two-part* annotation modes:
    - **One-part:** A single element (e.g., figure-caption inconsistency).
    - **Two-part:** Two separate elements (e.g., figure vs. text, or figure vs. figure).
- Specify for each part whether it was textual or visual:

---

[3]`https://openreview.net/forum?id=vXSCD3ToCS`

- – *Visual:* Select the page, then draw a bounding box on a rendered thumbnail version the PDF page.
- – *Textual:* Enter the page and line number, and copy the relevant text snippet from the PDF.
- Assign an inconsistency category via a drop-down menu.
- Provide a short free-text description of the inconsistency in their own words.

**Recorded Metadata.** Each annotation combined automatically and manually collected fields:

- **Automatically recorded:** element type (text or image), bounding box (relative coordinates) for visual selections, internal image identifier, reviewer's original comment.
- **Manually entered:** page and line numbers, copied textual content, inconsistency category, and a short description by the annotator.

**Example Output.** Annotations were stored in JSON format, combining visual/textual parts, reviewer comment, category, and description. We illustrate the example annotation output in JSON format in Fig. 8.

### D.3    STATISTICS OF INCONSISTENCY COLLECTION.

The annotation resulted in 384 inconsistencies from 353 ICLR papers. The average page count of each PDF was 16 pages. A total of 29 papers (7.6%) had more than one inconsistency. The paper subjects were equally distributed across the range of topics for ICLR[4], with (1) representation learning (26.0%), (2) transfer learning (10.4%), (3) generative models (9.6%) and datasets and benchmarks (7.8%) being the most frequent topics.

We identified 15 categories of inconsistencies based on the elements involved, with the distribution shown in Fig. 9. The most common cases were figure–text mismatches and intra-figure inconsistencies.

### D.4    LLM-BASED QUESTION GENERATION

#### D.4.1    INCONSISTENCY IDENTIFICATION (IDENT)

The *Inconsistency Identification (Ident)* task was the first benchmark task we designed. For each annotated inconsistency, we instructed *Gemini Flash* to generate a multiple-choice question (MCQ) with four options, one of which correctly describes the inconsistency.

**Inputs.** As input to the model, we provided:

- The annotated context (visual and/or textual parts).
- The annotator's free-text description of the inconsistency.

**Prompt.** After experimenting with several formulations, we found that a minimalist prompt yielded the most creative and plausible distractors. We provide the final version of the prompt in Fig. 10.

**Output Format.** The model produced a structured output containing the question, the correct answer, and three distractor answers.

**Manual Verification.** Each generated question underwent manual verification:

- **Correct answer:** must (1) faithfully reflect the annotator's description and (2) directly connect to the annotated context.
- **Distractors:** must (1) be grounded in the annotated inconsistency parts, (2) avoid obvious contradictions within the answer itself, (3) only mention elements present in the provided context, and (4) describe an inconsistency rather than confirming a correct fact from the paper

---

[4]https://iclr.cc/Conferences/2025/CallForPapers

We also post-processed the text to remove stylistic artifacts often appended by the LLM, e.g. the parenthetical *", indicating an inconsistency."*

**Example Output.** We illustrate an example of the generated multiple-choice question for inconsistency identification task in Fig. 11.

### D.4.2 DEBIASING THE INCONSISTENCY IDENTIFICATION TASK

**Initial Observations.** When first evaluating the *Ident* task with *Gemini 2.5 Flash*, we observed unexpectedly high accuracy:

- $84.4\%$ with the original LLM-generated questions. (e.g.: *"What inconsistency is observed between Figure 2 and the accompanying text regarding the generated road network?"*)
- $79.4\%$ after replacing the LLM-generated question with the generic formulation: *"What is the inconsistency in these parts of a scientific paper?"*

Even in a sanity check where the model was shown only the question and answer options (without the annotated context), performance remained at $57.6\%$ accuracy, far above the random baseline of $25\%$. This indicated strong reliance on linguistic cues in the answer phrasing.

**Mitigation Strategies.** Moving forward, we solely employed the generic question formulation throughout all inconsistencies. For reducing the without context accuracy $Acc_{nc}$, we systematically explored ways of reducing linguistic priors by rewriting the answer options:

- Normalizing answer length: $Acc_{nc} = 48.1\%$.
- Filtering for MCQs where the correct answer is shortest: $Acc_{nc} = 46.2\%$.
- Rephrasing distractors according to best practices in MCQ test design (Gierl et al., 2017): $Acc_{nc} = 41.6\%$.
- Shortening all answer options into nominal style: $Acc_{nc} = 38.2\%$.

While these interventions reduced bias, they did not remove it completely.

**Structured Representation: Evidence–Claim JSON.** As a more robust solution, we abandoned free-form natural language and introduced a structured, human-readable JSON representation that removes stylistic cues while preserving the semantic contradiction. The schema is:

```
{
  "letter": "A" | "B" | "C" | "D",
  "attribute": str,
  "claim": {
    "source": "expectation" | str,
    "statement": str
  },
  "evidence": {
    "source": str,
    "statement": str
  }
}
```

**Patterns.** Two patterns of contradiction are covered:

- **Claim vs. Evidence:** A claim from one paper element is contradicted by evidence from another.
- **Expectation vs. Evidence:** A claim contradicts common expectations of scientific correctness. In this case, the claim's source is always "expectation"

We prompted *Gemini 2.5 Flash* to convert the natural language MCQs into this structured format. The full prompt can be inspected in App. F.2. A 20% subset was manually validated for consistency.

**Effect on Model Behavior.** This representation further reduced the no-context accuracy to $34.0\%$. Given the full context, accuracy on the new JSON format decreased from $79.4\%$ to $69.5\%$. However, the fraction of performance attributable to visual grounding (Eq. 1) increased from $51.4\%$ to $53.8\%$. Thus, the structured format acts as a regularizer, forcing models to rely more strongly on the provided paper context.

**Example Output.** For the running example, we illustrate the example of debiased output in the evidence-claim JSON format for the inconsistency identification task in Fig. 12.

### D.4.3 INCONSISTENCY REMEDY TASK

**Task Design.** The *Inconsistency Remedy (Remedy)* task extends beyond identifying an inconsistency to determining how it can be resolved. To avoid linguistic artifacts, we directly employed a structured representation in JSON format. This representation adapts the Evidence–Claim schema to a more action-oriented form, the **Target–Action JSON**:

```
{
  "letter": "A" | "B" | "C" | "D",
  "attribute": str,
  "target": str,
  "other_involved": str,
  "action": "modify" | "remove" | "add" | "reposition" | "replace",
  "edit_statement": str,
  "reason": str
}
```

Here, attribute captures the element at issue, target specifies where the change is applied, other_involved records additional parts if necessary, and the fields action, edit_statement, and reason summarize the correction.

**LLM Conversion Process.** We found that prompting an LLM to directly convert the natural language MCQs from the *Ident* task into Target–Action JSON yielded the most reliable results in terms of readability and correctness. The prompt is depicted in Sec. F.3.

**Example Output.** Four our example used throughout this appendix, the task looks as follows: We illustrate the example output in Target-Action JSON format for the inconsistency remedy task in Fig. 13.

### D.4.4 INCONSISTENCY PAIR-MATCH TASK

**Task Design.** The *Inconsistency Pair-Match (Match)* task focuses on the subset of inconsistencies that involve two distinct visual parts. The model is presented with one element (text or visual) as the question context and must identify the corresponding inconsistent visual element among four options.

**Filtering of Eligible Cases.** Not all inconsistency categories are suitable for pair matching. Categories where the contradiction is contained entirely within a single element (i.e., *figure–caption*, *figure-only*, *table-only*, *table–caption*, *algorithm-only*) were excluded. This filtering left 135 out of the 262 inconsistencies in the dataset.

**Distractor Construction.** To ensure challenging and fair distractors, we extracted all figures, tables, and equations from the 242 papers in our dataset using *MinerU*[5], which produced image crops with unique IDs, modality labels, and page numbers. We then implemented a python script to sample distractors as follows:

- Distractors were restricted to the same modality as the correct answer.
- Preference was given to elements appearing on the same page or on adjacent pages to the correct element, so that distractors were topically similar.

---

[5]https://github.com/opendatalab/MinerU

- Sampling was done within the same paper. Each paper contained enough visual elements of the same modality so we didn't have to fallback to using elements from other papers.

This procedure reduced trivial elimination strategies (e.g., selecting "the only figure among tables") and forced models to consider fine-grained inconsistencies.

**Example Output.** In the running example, the annotated inconsistency links an in-line text with a figure. The text is fixed as the question context, and the answer options are image IDs referring to extracted figures, with one image ID being the correct image cropped in the annotations. We show the example of output used in the inconsistency pair-match task in Fig. 14.

# E  USER STUDY IMPLEMENTATION & STATISTICS

## E.1  IMPLEMENTATION OF THE USER STUDY

**Setup.** We conducted the user study in online form, using a custom web app. The participants were greeted with an onboarding screen, where they entered the following information to assess their eligibility to be included in the user study: (1) email address, (2) academic field, (3) academic level and(4) AI exposure. Afterwards, they were shown instructions for the survey and an introduction into the question formats and different context modalities. For each participants, ten tasks were randomly sampled from our dataset. For the first five tasks, the participants were shown the *Focused Context* with the exact cropped images and/or text passages from the paper. For the last five tasks, the participants were instructed to open a link to the original PDF and use the whole document to answer the question. In this case, they were provided with the visual element they should focus on in the paper. Screenshots of the user interface are provided in appendix H.

Upon submission, the following datapoints were saved automatically: (1) The task ID, (2) The chosen answer by the participants, (3) whether the task was correctly answered with/without context, (4) whether the question was accompanied by *Focused Context* or *Full Document Context* and (5) the time it took the participants to answer each question.

**Statistics.** Our eight participants all have a background in either artificial intelligence, computer science or mathematics at an academic level of PhD or higher (7 PhD, 1 postdoc). All stated to exhibit advanced exposure levels to AI, which we defined as being comfortable with reading, interpreting and critically evaluating AI scientific literature. The median answer time for questions without provided visual context was 45s, with *Focused Context* 145s and with *Whole Document Context* 169s. In 65% of the cases, participants changed their answers once provided with the context. In total, participants processed 80 inconsistencies.

## E.2  REPRESENTATIVENESS OF THE USER STUDY SUBSET

To ensure that the 40 samples used in the user study are representative of the full benchmark, we clarify that they were randomly drawn from the full dataset without any filtering or cherry-picking. As for some models, such as InternVL3.5 38B and Qwen2.5 VL 72B, accuracies on this subset were higher than their full-benchmark performance, we assessed whether this deviation was expected by running a 100-trial simulation. In each trial, we repeatedly sampled 40 random instances and computed the resulting scores.

As shown in Table 6, the simulation results indicate that the user-study subset happened to be in the upper quartile of model difficulty for these two models—a comparatively "easier" slice of the benchmark for them. Such variation is expected with small sample sizes and does not reflect systematic bias in the subset selection. Importantly, this strengthens our human-LMM comparison: even on a subset where LMMs performed better than their typical accuracy, humans still outperformed the models by a clear margin under both focused and whole-document conditions. If the subset had been closer to the mean model difficulty, the human-model gap would have been even larger. Thus, our user study results likely understate the core conclusion that humans substantially outperform current LMMs at detecting multimodal scientific inconsistencies.

Table 6: Model Performance Comparison: User Study vs. Simulation

| Model (Context) | User Study Subset (Reported) | Simulation Mean | Simulation Q3 (75th Percentile) | Human Perf. (Reference) |
|---|---|---|---|---|
| **InternVL 3.5 38B (Focused)** | 71.1% | 58.6% | 65.0% | 77.5% |
| **InternVL 3.5 38B (Whole Document)** | 40.5% | 30.6% | 35.0% | 65.0% |
| **Qwen 2.5 VL 72B (Focused)** | 65.8% | 50.5% | 55.0% | 77.5% |
| **Qwen 2.5 VL 72B (Whole Document)** | 48.6% | 35.9% | 40.0% | 65.0% |

# F  LLM PROMPTS

Here we provide the full prompts used to instruct the LLMs.

## F.1  LLM PROMPT FOR REVIEW FILTERING

We provide the prompt for LLM-based review filtering in Fig. 15. Given the reviewer's comment, the model is instructed in a chain-of-thought prompting manner to systematically analyze each desired characteristic of a visual inconsistency. Few-shot examples help clarify the output format.

## F.2  LLM PROMPT FOR CONVERTING INTO EVIDENCE-CLAIM FORMAT

We provide the prompt for LLM-assisted conversion of natural language answers (for the inconsistency identification task) into evidence-claim JSON format in Fig. 16. The evidence-claim JSON format is used as answer options in the inconsistency identification task. The structured JSON-based answer representation is for mitigating the language biases in multiple-choice evaluation.

## F.3  LLM PROMPT FOR CONVERTING INTO TARGET-ACTION FORMAT

We provide the prompt for LLM-assisted conversion of natural language answers (for the inconsistency identification task) into target-action JSON format in Fig. 17. Based on the question-answer pairs in inconsistency identification task, we generate question with answers for the inconsistency remedy task. The target-action JSON format is used as answer options in the inconsistency remedy task.

# G  SCREENSHOTS OF THE ANNOTATION APP

We show some examples of the interface of the annotation tool in Fig. 18 and Fig. 19.

# H  SCREENSHOTS OF THE SURVEY APP

We show some examples of the interface of the survey web interface in Fig. 20, Fig. 21 and Fig. 22.

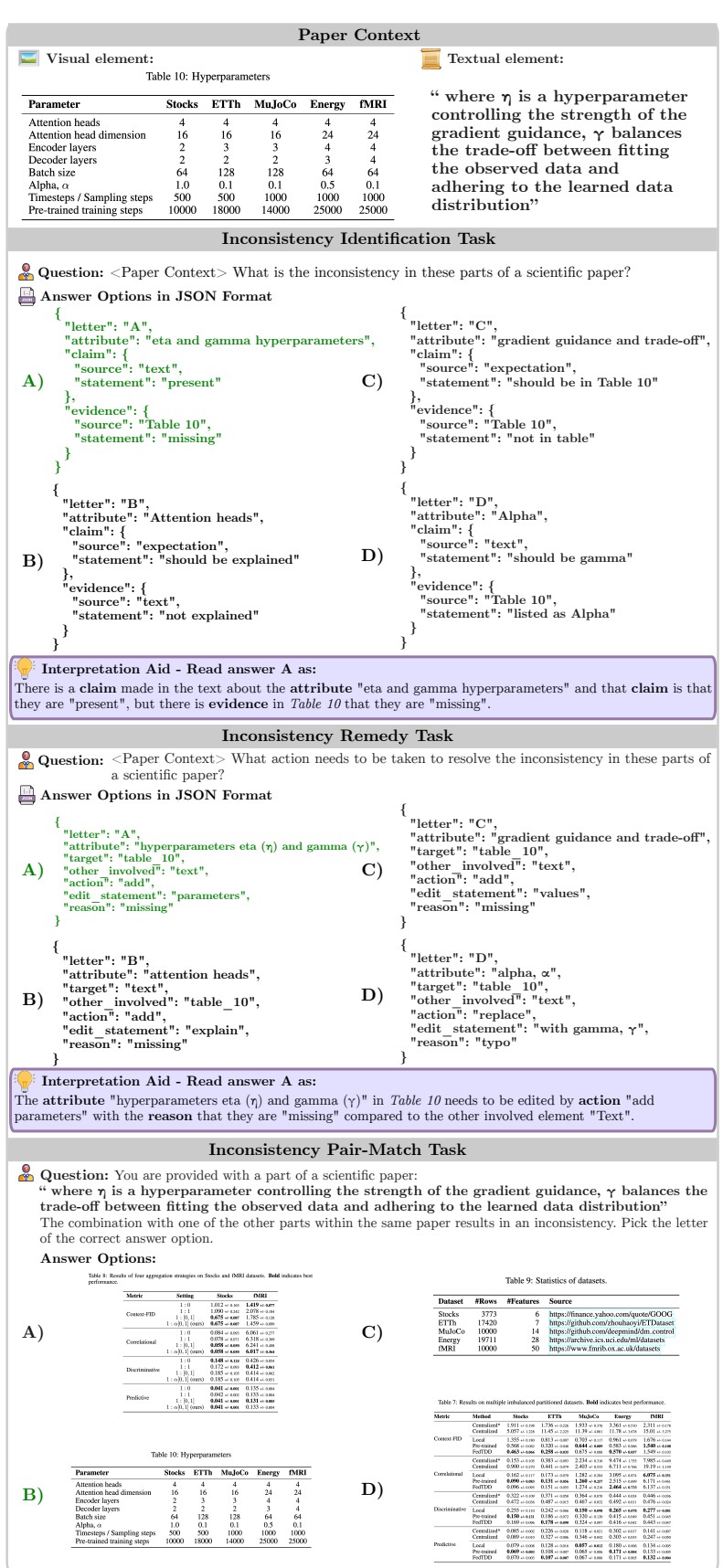

Figure 3: A qualitative example of a text–table inconsistency and its corresponding evaluation tasks of *Ident*, *Remedy* and *Match*.

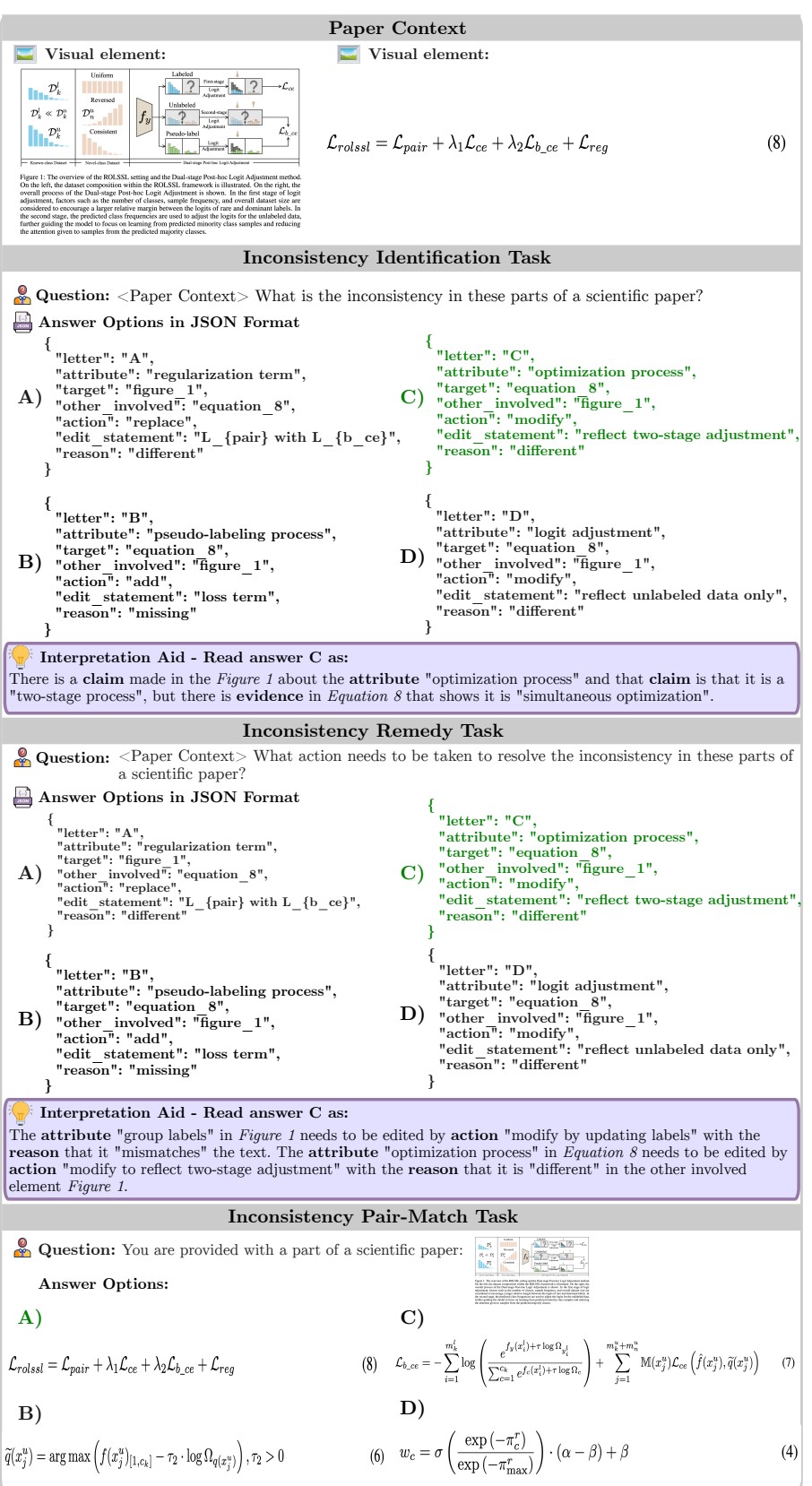

Figure 4: A qualitative example of a figure-equation inconsistency and its corresponding evaluation tasks of *Ident*, *Remedy* and *Match*.

---

**Full Reasoning Chain of InternVL 3.5 on inconsistency *3MDmM0rMPQ***

Okay, let's take a look at this question. The user is asking about inconsistencies in parts of a scientific paper based on the provided figure. The options are A to D, each pointing out a different potential issue.

First, I need to check each option against the figure provided. Let's start with option A: x-axis labels overlapping. Looking at the x-axis in Figure 6, the labels are "No Guardrail," "IPE," "OpenAI Moderation," "Perspective API," "Prompt-Based Llama2," and "LlamaGuard2." They seem to be spaced out and readable. There's no overlap here; each label is clearly visible. So A isn't the right answer.

Option B mentions the y-axis range being too narrow. The y-axis goes from 0 to 20.0, and the data points are all below 15, with error bars extending up to around 18. If the range was too narrow, the error bars might get cut off. But in the figure, the error bars are within the 0-20 range. The maximum error bar seems to reach just under 20, so the range is sufficient. B is not correct.

Option C states that error bars should be non-negative but extend below 0. Looking at the error bars, especially for "IPE" and "OpenAI Moderation," the error bars go below the 0 line. For example, the "No Guardrail" bar has an error bar that dips below 0 on the y-axis. Since error bars represent variability, they shouldn't go below zero if the metric (like jailbreaks) can't be negative. This seems like a valid inconsistency. So C is the answer.

Option D claims the legend isn't clear. The figure's legend uses different colors for each method, but the legend itself isn't shown in the image. However, the user's question refers to the figure's elements. Without a legend in the figure, that's an issue. But looking at the figure, each bar is labeled directly with the method name, so a legend isn't necessary. Therefore, D is not the issue here.

So, the most likely answer is C, because error bars extending below zero when the metric can't be negative is an inconsistency. The other options don't hold up upon closer inspection.

Figure 6: Raw reasoning output by InternVL3.5 38B on 3MDmM0rMPQ.

---

**Example Output After the LLM-assisted Filtering**

```
{
 "vXSCD3ToCS": {
   "has_inconsistency": true,
   "inconsistencies": [
     "Table 6: The performance improvement from using 20 years of data (MAE of 20.24) compared to
1 year (MAE of 21.90) in the 3-day setting is marginal, contradicting the emphasis on the scale of
DynST as a major contribution.",
     "Figure 2: The bottom left corner shows a road segment between two points that is not
represented as an edge in the topology, which appears inconsistent with the actual road network.",
     "Figure 5: The description of the figure suggests changes in sensors deployment, not actual road
network dynamics, which contradicts the paper's claim of considering dynamic road network
topology."
   ]
 }
}
```

Figure 7: Example output after the LLM-assisted filtering.

**Example Annotation Output in JSON Format**

```
{
  "inconsistency_parts": [
    {
      "type": "image",
      "page": 5,
      "image_id": "vXSCD3ToCS_5_a1e8a4c6",
      "bbox": { "x": 0.50, "y": 0.25, "width": 0.35, "height": 0.31 }
    },
    {
      "type": "text",
      "page": 5,
      "line": 262,
      "content": "The results demonstrate ..."
    }
  ],
  "review_text": "Figure 2: The bottom left corner shows ...",
  "category": "figure-text",
  "description": "Missing edges between nodes compared to the claim."
}
```

Figure 8: Example annotation output in JSON format.

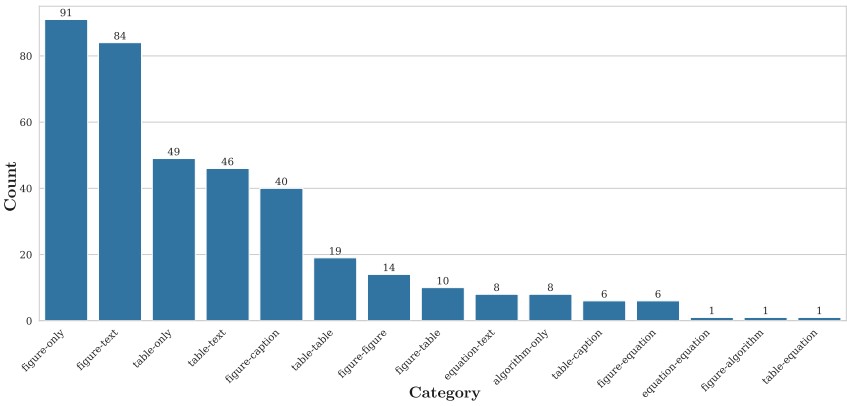

Figure 9: Distribution of inconsistency types. We identified 15 categories of inconsistencies based on the elements involved. The most common cases are figure-text mismatches and intra-figure (*figure-only*) inconsistencies.

**Prompt for Preparing MCQs in Inconsistency Identification Task**

You are a visual assistant that can analyze image and text excerpts from scientific papers. You receive either one image, two images, or a pair of image and text that contain a visual inconsistency flagged by reviewers. Alongside the content, you also receive a description of the inconsistency. Based on these, generate a multiple-choice question testing the model's ability to detect the inconsistency. Follow these strict rules:

- The question should directly reference the provided content of the paper.
- There must be exactly 4 answer choices.
- Only one answer should correctly describe the inconsistency.
- The 3 distractors must be plausible but incorrect. They should either be incorrect due to omission or subtle misinterpretations of the content.
- Do not invent details beyond what is provided.
- Clearly label the correct answer.

Figure 10: Prompt for preparing multiple-choice questions in the inconsistency identification task.

---

**Example of Generated MCQ in Inconsistency Identification Task**

```
{
  "mcq": {
    "default": {
      "question": "What inconsistency is observed between Figure 2 and the accompanying text
regarding the generated road network?",
      "correct": "The visualization in Figure 2 shows missing edges between nodes, which contradicts
the text's claim that the generated network perfectly matches the actual road network structure.",
      "incorrect": [
        "The visualization in Figure 2 shows extraneous edges between nodes, which contradicts the
text's claim that the generated network perfectly matches the actual road network structure.",
        "Figure 2 depicts only a disconnected portion of the road network, implying the algorithm did
not generate the complete structure.",
        "The blue nodes in Figure 2 are unevenly distributed, making it difficult to determine the
precise road paths."
      ],
      "letters": [
        "D", "A", "B", "C"
      ]
    }
  }
}
```

Figure 11: Example of generated multiple-choice question for the inconsistency identification task.

---

**Example of Debiased Output in Evidence-Claim JSON Format**

```
{
  "mcq": {
    "default": {
      "question": "What is the inconsistency in
these parts of a scientific paper?",

      "correct": {
        "letter": "A",
        "attribute": "edges",
        "claim": {
          "source": "text",
          "statement": "perfectly matches"
        },
        "evidence": {
          "source": "Figure 2",
          "statement": "missing edges"
        }
      },

      "incorrect": [
        {
          "letter": "C",
          "attribute": "edges",
          "claim": {
            "source": "text",
            "statement": "perfectly matches"
          },
          "evidence": {
            "source": "Figure 2",
            "statement": "extraneous edges"
          }
        },
        {
          "letter": "D",
          "attribute": "network",
          "claim": {
            "source": "expectation",
            "statement": "complete structure"
          },
          "evidence": {
            "source": "Figure 2",
            "statement": "disconnected portion"
          }
        },
        {
          "letter": "B",
          "attribute": "nodes",
          "claim": {
            "source": "expectation",
            "statement": "evenly distributed"
          },
          "evidence": {
            "source": "Figure 2",
            "statement": "unevenly distributed"
          }
        }
      ],
      "letters": [
        "A", "C", "D", "B"
      ]
    }
  }
}
```

Figure 12: Example of debiased output in evidence-claim JSON format for the inconsistency identification task.

---

**Example of Output in Target-Action JSON Format**

```
{
  "mcq": {                                          {
    "edit": {                                         "letter": "D",
      "question": "What action needs to be            "attribute": "road network",
taken to resolve the inconsistency in these parts     "target": "figure_2",
of a scientific paper?",                              "other_involved": "algorithm",
      "correct": {                                    "action": "modify",
        "letter": "A",                                "edit_statement": "disconnected portion",
        "attribute": "edges",                         "reason": "incomplete structure"
        "target": "figure_2",                       },
        "other_involved": "text",                   {
        "action": "add",                              "letter": "B",
        "edit_statement": "missing edges",            "attribute": "blue nodes",
        "reason": "contradicts claim"                 "target": "figure_2",
      },                                              "other_involved": null,
                                                      "action": "modify",
      "incorrect": [                                  "edit_statement": "distribute nodes evenly",
        {                                             "reason": "unclear paths"
          "letter": "C",                            }
          "attribute": "edges",                   ],
          "target": "figure_2",                   "letters": [
          "other_involved": "text",                 "A", "C", "D", "B"
          "action": "remove",                     ]
          "edit_statement": "extraneous edges",   }
          "reason": "contradicts claim"         }
        },                                      }
```

Figure 13: Example of output in target-action JSON format for the inconsistency remedy task, directly converted from the natural language MCQs from the inconsistency identification task.

---

**Example of Output for the Inconsistency Pair-Match Task**

```
{
  "mcq": {
    "part_pair": {
      "question": "The results demonstrate that the adjacency matrix generated by our algorithm
perfectly matches the actual road network structure.",
      "correct": "vXSCD3ToCS_5_a1e8a4c6",
      "incorrect": [
        "vXSCD3ToCS_5_image_figure3",
        "vXSCD3ToCS_5_image_figure4",
        "vXSCD3ToCS_6_image_figure5"
      ],
      "letters": [
        "D", "A", "C", "B"
      ]
    }
  }
}
```

Figure 14: Example output for the inconsistency pair-match task.

---

**Prompt for LLM-based Review Filtering**

You are an AI assistant specialized in analyzing academic paper reviews. Your task is to identify inconsistencies between visual elements (such as figures and tables) and their corresponding text descriptions in the original paper being reviewed. These inconsistencies should be explicitly mentioned or highlighted by the reviewer in their review.

Here is the paper review you need to analyze:

<review>

{prompt}

</review>

Instructions:

1. Carefully read through the entire review.

2. Focus exclusively on identifying instances where the reviewer mentions inconsistencies in the original paper between visual elements (figures, tables, graphs, etc.) and their corresponding text descriptions.

3. For each identified inconsistency:

    a. Determine the type of mismatch (e.g., figure legend vs. content, text results vs. figure data, table values vs. text mentions)

    b. Note the specific location or reference in the original paper (e.g., figure number, table number, page number if available)

    c. Briefly describe the nature of the inconsistency as mentioned by the reviewer

4. Disregard any general inconsistencies that are not related to vision-text mismatches in the original paper.

Before providing your final response, analyze the review in <review_analysis> tags:

5. List all mentions of visual elements in the review.

6. For each visual element, note whether the reviewer mentions any inconsistencies with the text.

7. For identified inconsistencies, write down the specific quote from the review that mentions it.

This analysis will help ensure a thorough examination of the review and prevent misinterpretation of inconsistencies within the review itself versus those in the original paper.

After your analysis, present your findings in JSON format. Each identified inconsistency should be an object in an array, with the following structure:

```
{
    "has_inconsistency": boolean,
    "inconsistencies": [
        "string (brief explanation of the inconsistency, always including the place in the original paper where it is
located and as close to the reviewer's text as possible)", // Additional inconsistencies...
    ]
}
```

If no vision-text inconsistencies in the original paper are mentioned by the reviewer,
return:

```
{
    "has_inconsistency": false,
    "inconsistencies": [ ]
}
```

Example of desired output structure (purely for format, not content):

```
{
    "has_inconsistency": true,
    "inconsistencies": [
        "Table 1: The performance for model A is 69.74 percent but the text mentions 65.47 percent.",
        "The text refers to Group 1 and Group 0, but Figure 1 labels the groups as Group 1 and Group 2."
    ]
}
```

Remember to focus solely on vision-text mismatches in the original paper as mentioned by the reviewer. Provide clear, concise descriptions that make it easy for researchers to locate and verify the inconsistencies in the original paper based on the review's comments.

---

Figure 15: Prompt for LLM-based review filtering.

**Prompt for Converting Natural Language Answer into Evidence-Claim JSON Format**

You are a system that converts multiple choice question answers into Evidence-Claim JSON format.

Evidence-Claim JSON format:
```json
{
  "letter": "A" | "B" | "C" | "D",
  "attribute": str,
  "claim": {
    "source": "expectation" | str,
    "statement": str
  },
  "evidence": {
    "source": str,
    "statement": str
  },
}
```

There are two patterns of answer options:

Pattern 1: One part of the answer makes a claim that is contradicted by evidence in another part

Example:
```json
{
  "letter": "C", // The letter of the answer option
  "attribute": "optimal trade-off", // The attribute in the center of the answer option (e.g. rank parameter, complexity, name, etc.)
  "claim": {
    "source": "caption", // The source the claim about the attribute is based on (e.g., caption, text, figure_1 etc.)
    "statement": "at 128 tokens" // A brief 2-3 words description
  },
  "evidence": {
    "source": "plot", // The source the evidence about the attribute contradicting the claim is based on (e.g., plot, table, equation_2 etc.)
    "statement": "not visible at 128 tokens" // A brief 2-3 words description
  },
}
```

Pattern 2: One part of the answer makes a claim that contradicts common expectations to scientific correctness

Example:
```json
{
  "letter": "A",
  "attribute": "legend",
  "claim": {
    "source": "expectation", // In that case, the source for claim is always "expectation"
    "statement": "shouldn't occlude plot"
  },
  "evidence": {
    "source": "figure_8",
    "statement": "occludes plot"
  },
}
```

Given:
- The question
- The answer options with letters (A, B, C, D)
- The correct answer letter
- The visual elements relevant to the inconsistency

Convert each multiple choice question answer (A, B, C, D) into the Target-Action JSON format. Ensure that the answer letters remain consistent with the input. Keep the JSON output concise. Do not use adjectives or any other descriptive language. The goal is to remove linguistic cues and focus on the core content of each answer option.

Figure 16: Prompt for LLM-assisted conversion of natural language answers of the inconsistency identification task into evidence-claim JSON format. The evidence-claim JSON format is used as answer options in the inconsistency identification task.

**Prompt for Converting Natural Language Answer into Target-Action JSON Format**

You are a system that converts multiple choice question answers about inconsistencies in scientific papers into Target-Action JSON format. The goal is to identify what needs to be changed in the paper to resolve the inconsistency.

Target-Action JSON format:
```json
{
  "letter": "A" | "B" | "C" | "D",
  "attribute": str, // the core element at issue (e.g., legend, methods evaluated, F1 scores)
  "target": str, // where the edit is applied (e.g., caption, figure_4b, table_5, equation_2)
  "other_involved": str // (optional) other elements involved in the inconsistency, comma-separated
  "action": "modify" | "remove" | "add" | "reposition" | "replace",
  "edit_statement": str, // short 2-3 words description of the needed change (exclude word from action)
  "reason": str // why the change is needed in 2-3 words
}
```

Example:
```json
{
  "letter": "C",
  "attribute": "windows",
  "target": "figure_1b",
  "other_involved": "figure_1a",
  "action": "modify",
  "edit_statement": "align door position",
  "reason": "different"
}
```

Given:
- The question
- The answer options with letters (A, B, C, D)
- The correct answer letter
- The visual elements relevant to the inconsistency

Convert each multiple choice question answer (A, B, C, D) into the Target-Action JSON format. Ensure that the answer letters remain consistent with the input. Keep the JSON output concise. Do not use adjectives or any other descriptive language. Most important is to remove linguistic cues and focus on the core content of each answer option.

Figure 17: Prompt for LLM-assisted conversion of natural language answers of the inconsistency identification task into target-action JSON format. The target-action JSON is used as answer options in the inconsistency remedy task.

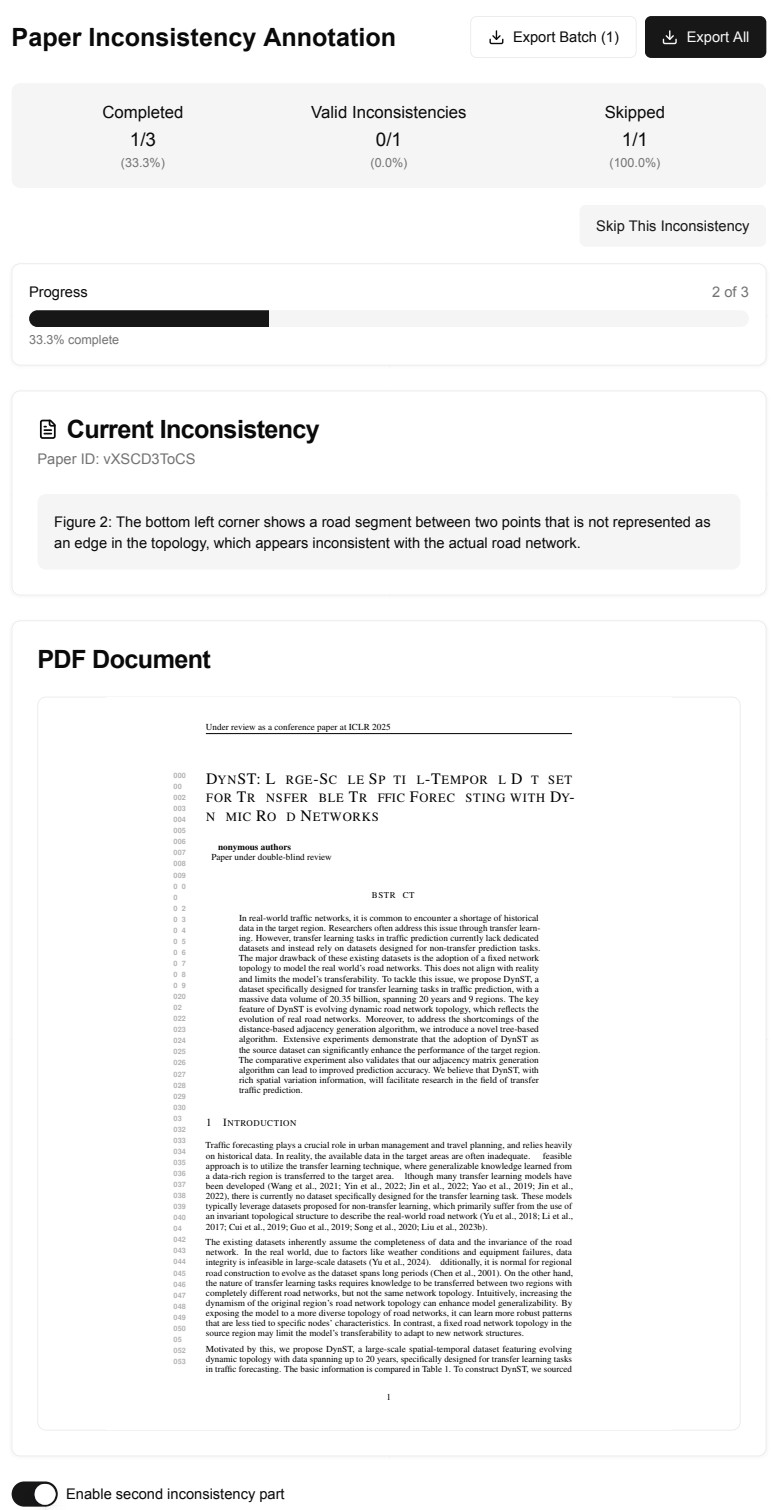

Figure 18: First part of annotation app showing an overview over the annotation progress and embedded original PDF file.

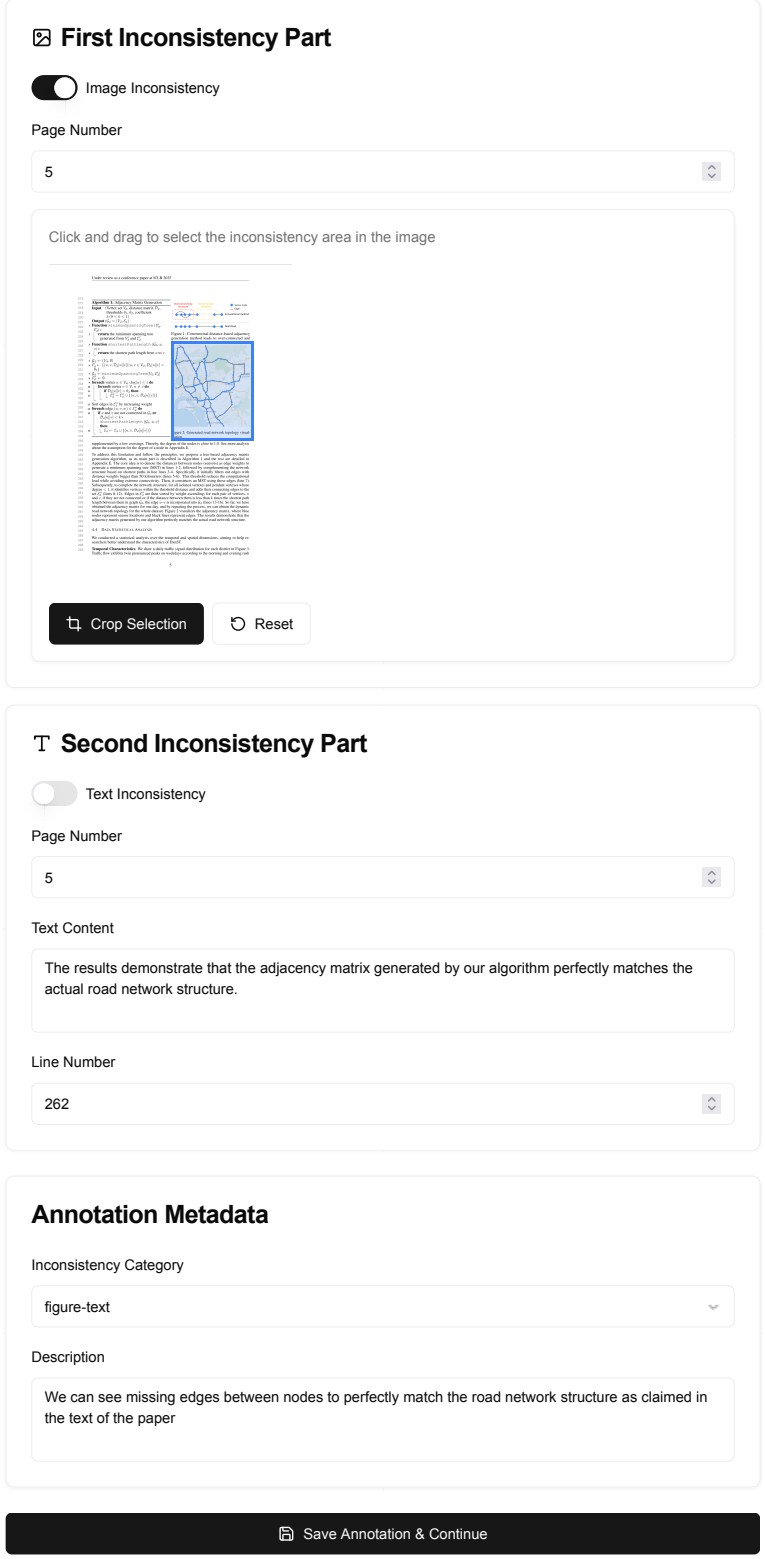

Figure 19: Second part of annotation app for drawing bounding boxes, entering text and further details about the inconsistency.

**Research Survey**

Question 1 of 10 - Phase 1

**Question (Phase 1: Initial Guess)**

What is the inconsistency in these parts of a scientific paper?

○ The 'Original Question' uses a continuous 1-10 scale, but the 'Converted Statements' only group responses into four discrete ranges.

○ The 'Original Question' defines 1 as 'Completely Dissatisfied' and 10 as 'Completely Satisfied', whereas the 'Converted Statements' redefine the lower numbers (1,2) as 'very satisfied' and the higher numbers (9,10) as 'very dissatisfied'.

○ The 'Original Question' allows for open-ended 'Unstructured Survey Questions', while the 'Converted Statements' are limited to 253 pre-defined value-expressing statements, indicating a change in data type.

○ The inconsistency lies in the 'LMs' Predictions' section, where Person A and Person B exhibit different accuracy percentages (56% vs 67%).

Submit Initial Guess

Figure 20: First part of survey interface showing a question with no context provided.

## Research Survey

Question 1 of 10 – Phase 2

**Context**

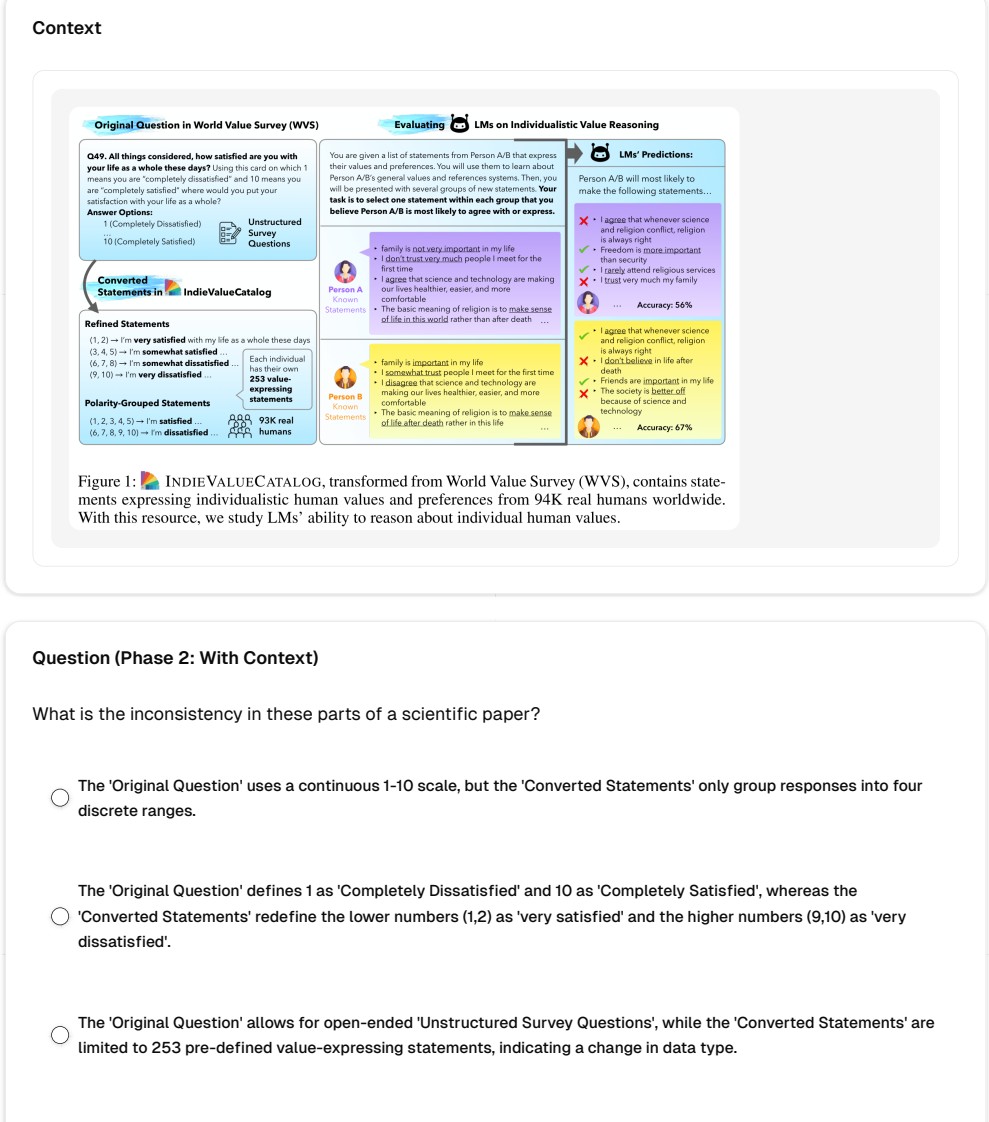

Figure 1: INDIEVALUECATALOG, transformed from World Value Survey (WVS), contains statements expressing individualistic human values and preferences from 94K real humans worldwide. With this resource, we study LMs' ability to reason about individual human values.

**Question (Phase 2: With Context)**

What is the inconsistency in these parts of a scientific paper?

○ The 'Original Question' uses a continuous 1-10 scale, but the 'Converted Statements' only group responses into four discrete ranges.

○ The 'Original Question' defines 1 as 'Completely Dissatisfied' and 10 as 'Completely Satisfied', whereas the 'Converted Statements' redefine the lower numbers (1,2) as 'very satisfied' and the higher numbers (9,10) as 'very dissatisfied'.

○ The 'Original Question' allows for open-ended 'Unstructured Survey Questions', while the 'Converted Statements' are limited to 253 pre-defined value-expressing statements, indicating a change in data type.

○ The inconsistency lies in the 'LMs' Predictions' section, where Person A and Person B exhibit different accuracy percentages (56% vs 67%).

Next Question

Figure 21: Second part of survey interface showing question with *Focused Context*.

**Research Survey**

Question 6 of 10 - Phase 2

**Context**

**PDF Document**

For this question, you can scroll the whole paper PDF to answer.

🔲 Open PDF Document

💡 **Pay attention to the following parts of the paper:**

**Text parts:** From Line 371

**Visual parts:** Figure 2

**Question (Phase 2: With Context)**

What is the inconsistency in these parts of a scientific paper?

○ The text claims that 'our model exhibits a more concentrated peak near Qdifference = 0', but Figure 2 shows 'normal_ours' having a main peak that is clearly more negative than CQL's main peak.

○ Figure 2 illustrates that both models have their primary Q_difference concentrations around positive values, which contradicts the text's statement that both models display a peak at negative Q_difference values.

○ The text states that CQL shows more spread in the positive Q_difference direction, indicating more frequent overestimations, but Figure 2 clearly depicts that 'normal_ours' (the proposed method) has a more extensive and pronounced presence in the positive Q_difference region compared to CQL.

○ The text mentions that both models have a 'long tail extending toward positive values', but Figure 2 indicates that neither 'normal_cql' nor 'normal_ours' show any data points in the positive Q_difference range.

Next Question

Figure 22: Third part of survey interface showing question with *Full Document Context*.

