# OpenReview forum: "PRISMM-Bench: A Benchmark of Peer-Review Grounded Multimodal Inconsistencies"
_ICLR.cc/2026/Conference — ICLR 2026 Poster_

### Official Review · Reviewer_gBzJ · 2025-10-24

**Soundness:** 2
**Presentation:** 3
**Contribution:** 2
**Rating:** 6
**Confidence:** 4

**Summary:**

This paper presents PRISMM-Bench, a new benchmark designed to evaluate large multimodal models (LMMs) on their capacity to detect, remedy, and reason about inconsistencies between multimodal content (text, figures, tables, and equations) in scientific papers. The authors construct the benchmark by mining real reviewer-flagged inconsistencies from ICLR submissions, refining this through LLM-based filtering and human annotation, resulting in 262 validated inconsistency cases across 242 papers. The benchmark includes three multiple-choice tasks (inconsistency identification, remedy, and pair match) and introduces a structured JSON-based answer format to minimize shortcut exploitation in model evaluation. Results from evaluating 21 LMMs demonstrate that the tasks remain highly challenging, even for state-of-the-art proprietary models.

**Strengths:**

1. Authentic Benchmark Construction: A key strength is the grounding of all inconsistency cases in real reviewer feedback from ICLR 2025 submissions, ensuring ecological validity. The paper moves away from synthetic or artificially injected errors, avoiding pitfalls of prior work.
2. Thorough Pipeline and Documentation: The multi-stage pipeline involving LLM filtering, human verification, and metadata annotation is robust and clearly described, supporting reproducibility.
3. Bias Mitigation: The use of structured JSON answer formats is a methodologically novel move to counteract answer-choice linguistic shortcuts.
4. Comprehensive Empirical Analysis: The experimental results cover 21 models, consider various model sizes and architectures, different input granularities, and ablation on answer format and context.

**Weaknesses:**

1. Limited Benchmark Scale and Generalizability: Despite claims of diversity, the dataset is relatively small (262 inconsistencies), and coverage is restricted to rejected or withdrawn ICLR 2025 submissions, all presumably from the AI/ML domain.
2. Potential for Shortcut Leakage Remains: The proposed JSON-based format mitigates, but does not fully obviate, the risk of answer-pattern exploitation. As seen in Table 2/3, model accuracy without context remains above chance. More analysis on possible residual shortcuts and the robustness of the JSON representation to adversarial answer construction would strengthen the remedy.

**Questions:**

1. Dataset Scale/Future Expansion: Are there specific plans or pilots for expanding PRISMM-Bench to domains beyond machine learning (e.g., other scientific fields) to address the current limitations in scope and domain diversity?
2. Inconsistency Existence Judgment: Most scientific papers may have no detectable inconsistencies, yet models first need to judge inconsistency existence before analysis. Currently, PRISMM-Bench only uses samples from papers with reviewer-flagged inconsistencies, and all three tasks’ answer options assume "inconsistencies definitely exist". Has the team considered supplementing it with samples from inconsistency-free papers and adding a "This section has no inconsistencies" option to multiple-choice tasks?

---

> ### Author Response · Authors · 2025-11-20
>
> # Part 1/3
> We thank the reviewer for the thoughtful and constructive feedback. We appreciate the recognition of our benchmark’s authenticity, grounding all cases in real peer-review reports, along with the acknowledgment of our robust multi-stage data collection pipeline, our JSON-based debiasing strategy for mitigating shortcut behavior, and our comprehensive empirical analysis across diverse LMM architectures and input settings. We address the reviewer’s concerns regarding dataset scale, domain scope, and potential residual shortcuts in detail below.
> ## Weakness 1: Scope and Scale of the Dataset
> We appreciate the reviewer’s concern and have carefully evaluated the scalability and generality of PRISMM-Bench during the rebuttal period.
>
> ### Expanded Dataset and Re-evaluation
> To test whether our conclusions hold beyond the original set of 262 instances, we extended our pipeline to a new source: ICLR 2024, and collected and human-verified an additional 122 inconsistencies. We then re-ran all 21 model evaluations on this expanded benchmark.
>
> - The core performance trends and model rankings remained stable, with the only minor change being GPT-5 (high-reasoning) narrowly overtaking Gemini 2.5 Pro.
> - This stability demonstrates that our original benchmark size is sufficient to capture the difficulty of the task, and that our data collection pipeline scales effectively to new papers.
> - We have already updated the paper on the OpenReview page with the expanded dataset statistics and the full set of re-evaluation results.
>
> These results empirically demonstrate that the benchmark is already robust at the current scale and that our data collection pipeline generalizes to new paper collections.
>
> ### Why ICLR?
> Our reliance on ICLR is driven by practical constraints of open peer-review availability.
> - ICLR is the only major venue that publicly releases full reviews for withdrawn papers, which is essential for recovering inconsistencies that still exist in the manuscript.
> - NeurIPS and ICML do not guarantee public reviews for rejected or withdrawn submissions, making it impossible to systematically collect reviewer-identified inconsistencies at scale.
> - We also focus specifically on rejected or withdrawn papers without a rebuttal, because reviewer-flagged inconsistencies in accepted papers are typically corrected in the camera-ready version, breaking the alignment between the flagged issues and the available PDF.
>
> Given these constraints, ICLR currently provides the only practical large-scale source for reliably reconstructing multimodal inconsistencies grounded in expert peer review.
>
> ## Question 1: Domain Coverage and Future Expansion
> We restrict our current benchmark to AI/ML domain because human verifiation requires domain expertise to interpret subtle technical inconsistencies. Extending to other scientific fields necessitates recruiting experts from those communities, which is feasible but beyond the scope of the initial release.
>
> We fully agree that expanding PRISMM-Bench beyond machine learning would broaden generalizability.
> - We are already exploring expansion to other open-review ecosystem, such as F1000Reseasrch [1], which uses a transparent post-publication peer-review model and covers biomedical disciplines. Its openly accessible reviews and stable document–review alignment make it a promising next venue for extending PRISMM-Bench to the biomedical domain.
> - We explicitly acknowledge domain scope as a limitation and view PRISMM-Bench as a foundation for a broader family of peer-review-grounded multimodal benchmarks.
>
> [1] https://f1000research.com

---

> > ### Author Response · Authors · 2025-11-20
> >
> > # Part 2/3
> >
> > ## Weakness 2: Residual Shortcuts & Robustness of JSON
> > Thank you for raising this point. We agree that mitigating shortcut evaluation is crucial for a fair MCQ benchmark. Below we clarify the extent to which JSON reduces bias, describe the alternative debiasing strategies we attempted, and present new adversarial stress tests conducted during the rebuttal period.
> >
> > ### Why JSON Was Necessary: Alternative NL Debiasing Strategies Were Insufficient
> > During development, we explored multiple natural-language debiasing methods. Two representative approaches are:
> > **(1) "Rule-based" Concept Swapping**
> > We extracted technical concepts from the context and constructed distractors by systematically controlled substitions. However, this produced unnatural phrasing and did not meaningfully reduce bias, e.g., Gemini 2.5 Flash still achieved 43.6% without context.
> >
> > **(2) Multi-Agent Debiasing Pipeline (4 agents, iterative)**
> > We developed a computationally heavy multi-agent system (Debiaser -> Validator -> Refiner -> Regenerator) aimed at neutralizing shortcut signals detectable by a "blind" validator. The roles of the agents were defined by extensive prompting:
> > - **Debiaser**: Rewrites all options to equalize the structure and length of the options
> > - **Validator**: Answers the MCQ output from the Debiaser without any further context. Outputs the guess, reasoning why the correct answer was chosen and the other refuted, and if a distractor was refuted based on linguistics or conent
> > - **Refiner**: If Validator guessed correctly based on linguistics, the Refiner rephrases the options to integrate the Validator's feedback
> > - **Distractor Regenerator**: If Validator guessed correctly based on content, this agent generates a new set of distractors using the context from the paper, the reviewer's comment, and the Validator's feedback
> >
> > Despite multiple refinement rounds, this approach yielded 38.5% without context, still far above random chance and worse than JSON.
> >
> > We also notice that when annotators edited natural-language options, stylistic cues reappeared. By contrast, editing values inside a JSON key-value structure did not introduce detectable bias.
> > These failures highlight that natural language inherently carries stylistic cues that LMMs can exploit. In the meanwhile, JSON suppresses such artifact as it is formal and compact.
> >
> > ### Adversarial Answer Constructions: Stress-Testing the JSON Format
> > To more directly test whether JSON is robust to adversarial distractor design, we conducted a follow-up experiment. Instead of generating distractors from the annotated inconsistency description, we prepare distractors by sampling correct JSON answers from other unrelated inconsistency samples across the entire dataset, and evaluate models without context.
> >
> > Results (Full Set)
> > | Model                | Accuracy (Sampled distractor) | Accuracy (Generated distractor)    |
> > | -------------------- | ------------------- | --- |
> > | InternVL 3.5 38B$^R$ | 25.2%               |  26.3%   |
> > | Ovis 2 34B           | 31.3%               |   32.1%  |
> > | Qwen 2.5 VL 72B        | 32.8%               |  34.7%   |
> > | Gemini 2.5 Pro$^R$        | 26.3%               |  37.8%   |
> >
> > We see that for open-weight models, adversarial distractors produce almost identical scores. This confirms that JSON reduces task-specific linguistic cues so effectively that even distractors drawn from unrelated samples do not change model behavior.
> >
> > Only for the strongest proprietary model (Gemini Pro) we observe a modest increase. This is expected as strong frontier models are known to detect and exploit extermely subtle statistical patterns, even when those patterns are semantically irrelevant (e.g., shortcut artifacts in LLaVA-Bench, MMStar, and MMLU-Pro). However, more importantly, the scores remains low, indicating that JSON dramatically reduces bias compared to natural language (57–71% w/o context).
> >
> > **Conclusion:** JSON does not introduce new biases via distractor construction, and for the vast majority of models, adversarial distractors do not increase shortcut exploitation. The small residual bias for the strongest proprietary model reflects model capability, not a flaw in the JSON representation.

---

> ### Author Response · Authors · 2025-11-20
>
> # Part 3/3
> ## Question 2: Inconsistency Existence Probing
> Thank you for raising this point. We intentionally did not include a "No inconsistency" answer option for two methodological reasons:
>
> 1. **Psychometric Validity**: Extensive research in educational assessment shows that “None of the above” / “No error” options undermine the diagnostic value of multiple-choice questions:
>     - They shift the cognitive process from identifying the correct answer to eliminating distractors, which changes the nature of the task being assessed.
>     - They inflate guessing behavior and introduce additional shortcut opportunities unrelated to the underlying skill.
>
> This is consistently documented in the test-design literature [1,2]. Since one of our core contributions is reducing shortcut biases, including a "Non inconsistency" option would introduce this artifact that we are trying to eliminate.
>
> 2. **Ground-Truth Non-exhaustiveness**: Our ground truth is derived from reviewer-flagged inconsistencies. In this case,
>     - A reviewer not flagging an issue does not mean the paper or section is free of inconsistencies. Many inconsistencies go entirely undetected during peer review.
>     - Therefore, creating "negative" samples from unflagged text would introduce false negatives, compromising benchmark validity.
>
> Constructing a reliable "no inconsistency" class would require *exhaustive* expert verification of every section of every paper which is far beyond what reviwers provide or what can be ensured at scale.
>
> [1] Caldwell DJ, Pate AN. Effects of question formats on student and item performance. Am J Pharm Educ. 2013 May 13;77(4):71. doi: 10.5688/ajpe77471. PMID: 23716739; PMCID: PMC3663625.
>
> [2] Gierl, M. J., Bulut, O., Guo, Q., & Zhang, X. (2017). Developing, Analyzing, and Using Distractors for Multiple-Choice Tests in Education: A Comprehensive Review. Review of Educational Research, 87(6), 1082-1116. https://doi.org/10.3102/0034654317726529 (Original work published 2017)

---

> ### Author Response · Authors · 2025-11-27
> **Gentle Reminder**
>
> Dear Reviewer gBzJ,
>
> We would like to kindly draw your attention to our detailed rebuttal and the additional experiments we conducted in response to your comments. We have aimed to thoroughly address each of your concerns, and the revisions are marked in the updated manuscript.
> If you have a moment to review our responses, we would greatly appreciate your feedback. If our clarifications resolve your concerns, we kindly ask you to consider updating your evaluation. We are, of course, happy to clarify anything further before the discussion period closes on December 3.
>
> Thank you again for your time and constructive comments.

---

### Official Review · Reviewer_KPqE · 2025-10-30

**Soundness:** 3
**Presentation:** 4
**Contribution:** 3
**Rating:** 6
**Confidence:** 3

**Summary:**

This paper introduces PRISMM-Bench, a benchmark for evaluating LMMs on multimodal inconsistencies in scientific papers. Its primary contribution is sourcing 262 inconsistencies from *real ICLR 2025 peer reviews* rather than synthetic data. The benchmark includes three tasks (Identification, Remedy, Pair Match) and proposes a novel JSON-based answer format to mitigate linguistic bias in MCQ evaluation. Experiments demonstrate that even state-of-the-art LMMs (e.g., Gemini 2.5 Pro) struggle (max 54.2% accuracy), proving the benchmark's difficulty.

**Strengths:**

* **High Originality and Authenticity:** The paper addresses the critical problem of multimodal inconsistency using real-world data sourced directly from expert peer reviews. This is a significant improvement over benchmarks based on synthetic errors.
* **Novel Debiasing Method:** The structured JSON answer format is an innovative and effective method to combat linguistic shortcuts in MCQ evaluations. The user study provides strong evidence for its necessity and utility.

**Weaknesses:**

* **Limited Scale and Domain:** The benchmark's primary flaw is its small scale (262 instances) and narrow scope (only AI papers from ICLR 2025). This severely limits its statistical power and generality as a benchmark.
* **Methodological Gaps:** The paper fails to report Inter-Annotator Agreement (IAA) for its human-verified dataset. This is a crucial omission that makes it difficult to assess the objectivity and quality of the annotations.

**Questions:**

See Weakness.

---

> ### Author Response · Authors · 2025-11-20
>
> # Part 1/2
>
> We thank the reviewer for the thoughtful and positive assessment of our work. We appreciate the recognition of our benchmark’s authenticity, grounding all cases in real peer-review reports, as well as the acknowledgment of our robust data-collection pipeline, novel JSON-based debiasing strategy, and comprehensive empirical evaluation across 21 LMMs. We address the reviewer’s concerns regarding dataset scale, domain scope, and annotation methodology in detail below.
> ## Weakness 1: Scope and Scale of the Dataset
> We appreciate the reviewer's concern and have taken it seriously. In the short rebuttal window, we invested substantial effort to assess the scalability and robustness of our benchmark.
>
> ### Expanded Dataset and Re-evaluation
> To test whether our conclusions hold beyond the original set of 262 instances, we extended our pipeline to a new source: ICLR 2024, and collected and human-verified an additional 122 inconsistencies. We then re-ran all 21 model evaluations on this expanded benchmark.
> The core performance trends and model rankings remained stable, with the only minor change being GPT-5 (high-reasoning) narrowly overtaking Gemini 2.5 Pro. This stability demonstrates that our original benchmark size is sufficient to capture the difficulty of the task, and that our data collection pipeline scales effectively to new papers.
>
> **We have already updated the paper on the OpenReview page with the expanded dataset statistics and the full set of re-evaluation results.**
>
> ### Justification of Review Source
> Our reliance on ICLR is driven by practical constraints of open peer-review availability.
> - ICLR is the only major venue that publicly releases full reviews for withdrawn papers, which is essential for recovering inconsistencies that still exist in the manuscript.
> - NeurIPS and ICML do not guarantee public reviews for rejected or withdrawn submissions, making it impossible to systematically collect reviewer-identified inconsistencies at scale.
> - We also focus specifically on rejected or withdrawn papers without a rebuttal, because reviewer-flagged inconsistencies in accepted papers are typically corrected in the camera-ready version, breaking the alignment between the flagged issues and the available PDF.
>
> ### Domain Coverage and Future Expansion
> We restrict our current benchmark to the AI/ML domain because human verification requires domain expertise to interpret subtle technical inconsistencies. Extending to other scientific fields necessitates recruiting experts from those communities, which is feasible but beyond the scope of the initial release.
>
> We fully agree that the broader domain coverage is important. We are already exploring expansion to other open-review ecosystems, such as F1000Research [1], which uses a transparent post-publication peer-review model and covers biomedical disciplines. We explicitly acknowledge this as a limitation and regard our PRISMM-Bench as a first step towards broader, peer-review-grounded multimodal evaluation.
>
> [1] https://f1000research.com

---

> ### Author Response · Authors · 2025-11-20
>
> # Part 2/2
>
> ## Weakness 2: Methodological Gaps: Inter-Annotator Agreement (IAA)
> Thank you for highlighting this. We have addressed this omission and conducted an IAA study during the rebuttal period.
>
> ### Clarifying the Annotation Task.
> Our annotators do not generate inconsistencies themselves. Instead, they verify reviewer-flagged issues by checking whether the described inconsistency is present in the PDF, following a strict, rule-based protocol. This is an objective grounding task rather than a subjective interpretive one, which already limits annotator variance.
>
> ### IAA Study Setup
> To quantify reliability, we asked a senior researcher to re-annotate 32 randomly sampled instances from 10 papers. We evaluate agreement on two independent dimensions:
> 1. **Detection**: Whether the reviwer's comment corresponds to a real multimodal inconsistency in the PDF (Yes/No)
> 2. **Category Assignment**: Selecting one of the 15 predefined inconsistency types (e.g. figure-table, figure-caption).
>
> We estimate 95% confidence intervals via boostrap (1000 iterations) and report the **Inter-Annotator Agreement (IAA) Analysis ($N=32$)**
> | Metric | Inconsistency Detection | Category Assignment |
> | :--- | :--- | :--- |
> | **Agreement** | **93.75%** | **70.00%** |
> | **Cohen's Kappa ($\kappa$)** | **0.861** [CI: 0.604, 1.000] | **0.655** [CI: 0.286, 1.000] |
> | **Krippendorff's Alpha ($\alpha$)** | **0.864** | **0.669** |
> | **Interpretation** | *Almost Perfect Agreement* | *Substantial Agreement* |
>
> As shown in the table, detection, which is the component directly used in the benchmark construction, shows *Almost Perfect* agreement (Landis & Koch scale [2]). Category assignment shows *Substantial Agreement* which is expected given 15 fine-grained labels. Importantly, categories are not used in model evaluation; they are only descriptive metadata for dataset analysis.
>
> The results confirm that our annotation process is both reliable and objective.
>
> [2] Landis JR, Koch GG. The measurement of observer agreement for categorical data. Biometrics. 1977 Mar;33(1):159-74. PMID: 843571.

---

> ### Author Response · Authors · 2025-11-27
> **Gentle Reminder**
>
> Dear Reviewer KPqE,
>
> We would like to kindly draw your attention to our detailed rebuttal and the additional experiments we conducted in response to your comments. We have aimed to thoroughly address each of your concerns, and the revisions are marked in the updated manuscript.
> If you have a moment to review our responses, we would greatly appreciate your feedback. If our clarifications resolve your concerns, we kindly ask you to consider updating your evaluation. We are, of course, happy to clarify anything further before the discussion period closes on December 3.
>
> Thank you again for your time and constructive comments.

---

### Official Review · Reviewer_CmSk · 2025-10-31

**Soundness:** 3
**Presentation:** 3
**Contribution:** 3
**Rating:** 6
**Confidence:** 4

**Summary:**

This paper presents PRISMM-Bench, a benchmark for evaluating large multimodal models (LMMs) on identifying and fixing inconsistencies in research papers. The inconsistencies are sourced from those explicitly mentioned by reviewers in OpenReview reviews of ICLR 2025 papers. The benchmark consists of 3 tasks in multiple-choice format for 1) identifying the inconsistency (Ident),  proposing solution to fix inconsistency (Remedy), and matching elements of inconsistency (Match). Observing that LMMs tends to leverage text priors as shortcuts that heavily biased the results, the authors propose a debiasing method that converts the natural language options into JSON format which has shown to be effective. Experiment results of 21 LMMs on the benchmark reveals that even the best performing models struggles in these tasks especially in more realisitic long-context setting, showcasing the challenges of applying these LMMs as scholar assistants for tasks like identifying and fixing inconsistency with fine-grained grounding in multimodal context.

**Strengths:**

1. The presented task is both novel and of potential practical value. Testing LMMs capability in identifying and fixing inconsistency in papers could be a good indicator of their capabilities in fine-grained multimodal grounding and understanding, while also foster the development of tools based on these models to help authors check their papers.
2. The observation & analysis of choice-only shortcuts is insightful, and the proposed JSON-based debiasing method largely increase the utility of the benchmark and also provide a potentially useful debiasing approach for future MCQ-based benchmark construction.

**Weaknesses:**

1. For both the identification & remedy tasks, I am not fully convinced that the MCQ setup is the optimal way of presenting this task. While the MCQ task does help probe LMMs capabilities, they are both indirect and less realistic - in real world scenarios we would expect the LMMs to identify and remedy these inconsistencies from scratch instead of selecting from a set of options. Plus all the shortcut biases the authors have observed for MCQ. So I would consider formulate them as generation tasks that directly ask LMMs to output what are the inconsistencies & solutions. It seems to me the evaluation should also be easy and less biased - I would expect LLM-as-a-judge can determine if the generated inconsistency & solution are the same with human annotated ground-truth at a pretty high accuracy.
2. The subset results for the user study might not faithfully reflects the human-LMMs gap. In table 2, the JSON performance of InternVL3.5 38B R & Qwen 2.5 VL 72B on both focused & Document are significantly higher than their corresponding performance on the full set as shown in table 1.

**Questions:**

1. In addition to the weaknesses mentioned above, can the authors explain why the yield rate in terms of  # of human validated inconsistencies / # LLM filtered inconsistencies is so low? Judging from the numbers in the paper, only ~5% of the LLM recognized potential inconsistencies are validated, which significantly constraint the overall size of the benchmark.

---

> ### Author Response · Authors · 2025-11-20
>
> # Part 1/2
>
> We thank the reviewer for the thoughtful and constructive assessment of our work. We appreciate the reviewer’s recognition of the key strengths of PRISMM-Bench on the novelty and difficulty of evaluating LMMs on real multimodal inconsistencies, the value of leveraging OpenReview as an authentic source of expert-flagged errors, and the robustness of our multi-stage data‐collection pipeline and experimental analysis. We are encouraged that the reviewer views the task as practically meaningful and the methodology as insightful. Below, we address the concerns regarding the evaluation format and benchmark scope in detail.
>
> ## Weakness 1: Validity of Multiple-Choice vs. Open-Ended Evaluation
> Thank you for raising this important concern. We fully agree that open-ended generation is closer to real-world usage. Our choice of MCQ format was motivated by the need for a reliable, reproducible and bias-controlled evaluation protocol for benchmarking. To directly address your point, we conducted additional experiments during the rebuttal period using open-ended, free-form answers evaluated by LLM-as-a-judge.
>
> ### Open-Ended Evaluation Experiment
> We tested a representative subset of the identification task $(N=50)$ using open-ended responses from eight models. Their outputs were rated by three different judges (Gemini 2.5 Pro, GPT-5(high) and GLM 4.5V 106B) on a Likert-5 scale measuring semantic alignment with the ground-truth inconsistency. Each judge was run without and with Focused context.
>
> ### Results
> The results are the average Likert-scale score; the numbers in parentheses indicate the percentage of responses with a score greater than 3. R means rank number.
> | Candidate Model | Gemini 2.5 Pro Judge (w/o context) | | Gemini 2.5 Pro Judge (context) | | GPT-5 Judge (w/o context) | | GPT-5 Judge (context) | | GLM 4.5V 106B Judge (w/o context) | | GLM 4.5V Judge (context) | |
> |:---|:---|:---|:---|:---|:---|:---|:---|:---|:---|:---|:---|:---|
> | | **Score** | **R** | **Score** | **R** | **Score** | **R** | **Score** | **R** | **Score** | **R** | **Score** | **R** |
> | Gemini Pro 2.5$^R$ | 2.80 (44.0%) | 2 | 3.06 (54.0%) | 1 | 2.71 (40.0%) | 2 | 2.80 (44.0%) | 2 | 3.06 (54.0%) | 1 | 3.50 (62.0%) | 1 |
> | GPT-5 (high)$^R$ | 3.00 (54.0%) | 1 | 2.70 (42.0%) | 2 | 2.76 (52.5%) | 1 | 2.94 (52.0%) | 1 | 3.02 (51.0%) | 2 | 3.32 (58.0%) | 2 |
> | InternVL 3.5 38B$^R$ | 2.22 (34.0%) | 6 | 2.26 (34.0%) | 4 | 2.06 (25.0%) | 4 | 1.98 (24.4%) | 4 | 2.68 (48.0%) | 3 | 2.84 (46.0%) | 3 |
> | Qwen 2.5 VL 72B | 2.46 (38.0%) | 3 | 2.40 (38.0%) | 3 | 2.30 (28.3%) | 3 | 2.13 (26.7%) | 3 | 2.42 (34.0%) | 6 | 2.60 (40.0%) | 6 |
> | GLM 4.5V 106B$^R$ | 2.40 (40.0%) | 4 | 2.08 (28.0%) | 5 | 2.04 (22.0%) | 5 | 1.94 (22.0%) | 5 | 2.56 (42.0%) | 5 | 2.83 (47.9%) | 4 |
> | InternVL 3.5 8B$^R$ | 2.04 (28.0%) | 7 | 1.94 (28.0%) | 6 | 1.94 (18.0%) | 7 | 1.66 (18%) | 8 | 2.60 (42.0%) | 4 | 2.78 (42.0%) | 5 |
> | Gemma 3 12B | 2.24 (34.0%) | 5 | 1.78 (24.0%) | 8 | 2.03 (19.3%) | 6 | 1.78 (18.4%) | 6 | 2.22 (32.0%) | 7 | 2.50 (36.0%) | 7 |
> | Ovis 2 34B | 1.96 (20.0%) | 8 | 1.86 (22.0%) | 7 | 1.83 (14.0%) | 8 | 1.74 (16.0%) | 7 | 2.18 (32.0%) | 8 | 2.30 (32.0%) | 8 |
>
> ### Findings from the Open-Ended Evaluation
> **(1) LLM judges produced inconsistent scores across judges**
> Even with the same model outputs and same input context
> - Gemini 2.5 Pro and GPT-5 often disagreed on absolute scores and relative rankings.
> - Some models changed rank by 2-3 positions depending on the judge used.
> - GLM 4.5V 106B was particularly unstable as a judge.
> This variability undermines reproducibility.
>
> **(2) LLM judges produced inconsistent scores across runs**
> For proprietary reasoning models like GPT-5, `temperature=0` is not available, this means repeated evaluations of the same answers do not yield deterministic scores, and ranking variation could appear across repeated trials.
> This introduces noise that is unsuitable for benchmarking, especially when differences between strong models are often within 2-5 percentage points.
>
> **(3) Judge bias interacts with the model under evaluation**
> We observed some systematic patterns. Gemini tends to rate Gemini-style outputs more favorably. GPT-5 tends to rate GPT-family outputs higher, even when judged on identical content. This well-known "judge-model similarity bias" is documented in MT Bench [1], AlpacaEval [2] and other LLM-judge evaluations.
>
> [1] Zheng et al. Judging LLM-as-a-Judge with MT-Bench and Chatbot Arena.
>
> [2] Dubois et al. Length-Controlled AlpacaEval: A Simple Way to Debias Automatic Evaluators.

---

> > ### Author Response · Authors · 2025-11-20
> >
> > # Part 2/2
> >
> > ## Continuation of Weakness 1
> > As a conclusion, our additional open-ended evaluation shows promise for qualitative error analysis but also reveals judge inconsistency, judge-model bias, run-to-run variability and difficulty in replicating results.
> > ### Reasoning for Choice of MCQ
> > In comparison to open-ended evaluation, MCQ remains more reproducible. It provides deterministic scoring, controlled distractor difficulty, identifcal evaluation conditions for all models and no dependence on third-party proprietary judges.
> > Furthermore, our JSON-based debiasing removes the dominant shortcut patterns and brings model behavior closer to human reasoning (demonstrated via our user study).
> >
> > Therefore, we believe the MCQ format remains the more reliable and reproducible framework for PRISMM-Bench. We will include the open-ended results and insights in the camera-ready version to facilitate future exploration of hybrid evaluation protocols.
> >
> > ## Weakness 2: User Study Representativeness
> > We appreciate the reviewer's careful examination of the user study subset. We want to first clarify that the 40 samples used in the user study were randomly drawn from the full dataset, without any filtering or cherry-picking.
> > It is true that for two models, InternVL3.5 38B and Qwen2.5 VL 72B, their accuracies on this subset were higher than their full-benchmark performance. To assess whether this deviation was expected, we ran a 100-trial simulation, repeatedly sampling 40 random instances and computing the resulting scores. The results are shown below:
> >
> > | Model (Context) | User Study Subset (Reported) | Simulation Mean | Simulation Q3 (75th Percentile) | Human Perf. (Reference) |
> > | :--- | :---: | :---: | :---: | :---: |
> > | **InternVL 3.5 38B (Focused)** | 71.1% | 58.6% | 65.0% | 77.5% |
> > | **InternVL 3.5 38B (Whole Document)** | 40.5% | 30.6% | 35.0% | 65.0% |
> >  | **Qwen 2.5 VL 72B (Focused)** | 65.8% | 50.5% | 55.0% | 77.5% |
> > | **Qwen 2.5 VL 72B (Whole Document)** | 48.6% | 35.9% | 40.0% | 65.0% |
> >
> > ### Interpretation
> > This simulation shows that the user-study subset happened to be in the upper quartile of model difficulty for these two models, i.e., a comparatively "easier" slice of the benchmark for them. This is expected with small sample sizes and does not reflect systematic bias in how the subset was chosen.
> > More importantly, this strengthens our human-LMM comparison:
> > - Even on a subset where LMMs performed better than their typical accuracy, humans still outperformed the models by a clear margin under both focused and whole-document conditions.
> > - If the subset had been closer to the mean model difficulty (as our simulation indicate), the human-model gap would have been even larger.
> >
> > In this case, our user study still supports, and in fact slightly understates, the core conclusion: Humans substantially outperform current LMMs at detecting multimodal scientifc inconsistencies.
> >
> > ## Question 1: Low Yield Rate of Inconsistencies
> > Thank you for raising this question. The low yield rate is an expected consequence of intentionally designing the LLM-filtering stage for high recall (fewer false negatives) rather than high precision (fewer false positives). Missing a real inconsistency (a false negative) is far more costly than allowing more false positives, because
> > - True inconsistencies are rare relative to the total volume of OpenReview comments
> > - False negatives cannot be recovered later, while
> > - False positives are quick and inexpensive to refute during manual verification, since an annotator can immediately inspect the referenced region in our annotation tool.
> >
> > This design choice increases the number of LLM-flagged candidates but ensures that we do not miss the true reviewer-identified inconsistencies the benchmark aims to capture.
> >
> > There are two additional structural reasons behind the low yield rate:
> > ### 1. Inconsistencies due to omission
> > Many reviewer comments refer to missing content (e.g., "The text references Figure 3, but the figure is not present.")
> > Because our benchmark evaluates inconsistencies grounded in existing multimodal content, we cannot annotate cases where the relevant figure/table/equation does not exist in the PDF. These must be filtered out.
> > ### 2. External cross-paper inconsistencies
> > Reviewers frequently flag issues involving comparisons to external papers (e.g., incorrect baseline numbers copied from prior work).
> > Since PRISMM-Bench focuses on within-document multimodal inconsistencies, we exclude inconsistencies that require external documents to verify. **This is an interesting direction, and highlights it as promising future work.**
> >
> > Overall, the low yield rate reflects (1) a delibrate high-recall filtering strategy and (2) systematic exclusion of categories outside the scope of the benchmark, rather than shortcomings in the annotation process.

---

> ### Author Response · Authors · 2025-11-27
> **Gentle Reminder**
>
> Dear Reviewer CmSk,
>
> We would like to kindly draw your attention to our detailed rebuttal and the additional experiments we conducted in response to your comments. We have aimed to thoroughly address each of your concerns, and the revisions are marked in the updated manuscript.
> If you have a moment to review our responses, we would greatly appreciate your feedback. If our clarifications resolve your concerns, we kindly ask you to consider updating your evaluation. We are, of course, happy to clarify anything further before the discussion period closes on December 3.
>
> Thank you again for your time and constructive comments.

---

### Official Review · Reviewer_stE4 · 2025-11-01

**Soundness:** 3
**Presentation:** 3
**Contribution:** 3
**Rating:** 6
**Confidence:** 4

**Summary:**

This paper introduces PRISMM-Bench, the first benchmark derived from real peer-reviewed inconsistencies in multimodal scientific papers. It systematically mines OpenReview comments to construct tasks that assess LMMs on detecting and correcting inconsistencies across text, figures, tables, and equations. The work fills a meaningful gap in multimodal reasoning evaluation in terms of inconsistency detection in scientific papers.

**Strengths:**

1. The topic of instructing LMMs to discover the inconsistencies is interesting. This is a very difficult task, even for reasoning models, and it evaluates various abilities, including reasoning, scientific graph understanding, and long multimodal context understanding.

2. The idea of using reviews from the OpenReview website is clever, which reflects the real-world scenarios and leverages the implicit human-labeled data.

3. The data collection pipeline is reasonable, and the authors have done extensive experiments to provide insightful conclusions.

**Weaknesses:**

1. The questions are all in multiple-choice form. Although this is easy to evaluate, it is not representative of how LMMs will be used in real-world scenarios, which may weaken the results and conclusions of the paper.

2. The scope of the benchmark is another concern. The benchmark only contains 262 inconsistencies, which may be too small to derive a solid conclusion. Besides, the paper is only from ICLR 2025, which mainly consists of AI fields. The LMMs' ability to discern inconsistencies in other fields has not been explored.

**Questions:**

Please refer to the weakness part.

---

> ### Author Response · Authors · 2025-11-20
>
> # Part 1/2
>
> We thank the reviewer for the thoughtful and constructive assessment of our work. We appreciate the reviewer’s recognition of the key strengths of PRISMM-Bench on the novelty and difficulty of evaluating LMMs on real multimodal inconsistencies, the value of leveraging OpenReview as an authentic source of expert-flagged errors, and the robustness of our multi-stage data‐collection pipeline and experimental analysis. We are encouraged that the reviewer views the task as practically meaningful and the methodology as insightful. Below, we address the concerns regarding the evaluation format and benchmark scope in detail.
>
> ## Weakness 1: Validity of Multiple-Choice vs. Open-Ended Evaluation
> Thank you for raising this important concern. We fully agree that open-ended generation is closer to real-world usage. Our choice of MCQ format was motivated by the need for a reliable, reproducible and bias-controlled evaluation protocol for benchmarking. To directly address your point, we conducted additional experiments during the rebuttal period using open-ended, free-form answers evaluated by LLM-as-a-judge.
>
> ### Open-Ended Evaluation Experiment
> We tested a representative subset of the identification task $(N=50)$ using open-ended responses from eight models. Their outputs were rated by three different judges (Gemini 2.5 Pro, GPT-5(high) and GLM 4.5V 106B) on a Likert-5 scale measuring semantic alignment with the ground-truth inconsistency. Each judge was run without and with Focused context.
>
> ### Results
> The results are the average Likert-scale score; the numbers in parentheses indicate the percentage of responses with a score greater than 3. R means rank number.
> | Candidate Model | Gemini 2.5 Pro Judge (w/o context) | | Gemini 2.5 Pro Judge (context) | | GPT-5 Judge (w/o context) | | GPT-5 Judge (context) | | GLM 4.5V 106B Judge (w/o context) | | GLM 4.5V Judge (context) | |
> |:---|:---|:---|:---|:---|:---|:---|:---|:---|:---|:---|:---|:---|
> | | **Score** | **R** | **Score** | **R** | **Score** | **R** | **Score** | **R** | **Score** | **R** | **Score** | **R** |
> | Gemini Pro 2.5$^R$ | 2.80 (44.0%) | 2 | 3.06 (54.0%) | 1 | 2.71 (40.0%) | 2 | 2.80 (44.0%) | 2 | 3.06 (54.0%) | 1 | 3.50 (62.0%) | 1 |
> | GPT-5 (high)$^R$ | 3.00 (54.0%) | 1 | 2.70 (42.0%) | 2 | 2.76 (52.5%) | 1 | 2.94 (52.0%) | 1 | 3.02 (51.0%) | 2 | 3.32 (58.0%) | 2 |
> | InternVL 3.5 38B$^R$ | 2.22 (34.0%) | 6 | 2.26 (34.0%) | 4 | 2.06 (25.0%) | 4 | 1.98 (24.4%) | 4 | 2.68 (48.0%) | 3 | 2.84 (46.0%) | 3 |
> | Qwen 2.5 VL 72B | 2.46 (38.0%) | 3 | 2.40 (38.0%) | 3 | 2.30 (28.3%) | 3 | 2.13 (26.7%) | 3 | 2.42 (34.0%) | 6 | 2.60 (40.0%) | 6 |
> | GLM 4.5V 106B$^R$ | 2.40 (40.0%) | 4 | 2.08 (28.0%) | 5 | 2.04 (22.0%) | 5 | 1.94 (22.0%) | 5 | 2.56 (42.0%) | 5 | 2.83 (47.9%) | 4 |
> | InternVL 3.5 8B$^R$ | 2.04 (28.0%) | 7 | 1.94 (28.0%) | 6 | 1.94 (18.0%) | 7 | 1.66 (18%) | 8 | 2.60 (42.0%) | 4 | 2.78 (42.0%) | 5 |
> | Gemma 3 12B | 2.24 (34.0%) | 5 | 1.78 (24.0%) | 8 | 2.03 (19.3%) | 6 | 1.78 (18.4%) | 6 | 2.22 (32.0%) | 7 | 2.50 (36.0%) | 7 |
> | Ovis 2 34B | 1.96 (20.0%) | 8 | 1.86 (22.0%) | 7 | 1.83 (14.0%) | 8 | 1.74 (16.0%) | 7 | 2.18 (32.0%) | 8 | 2.30 (32.0%) | 8 |
>
> ### Findings from the Open-Ended Evaluation
> **(1) LLM judges produced inconsistent scores across judges**
> Even with the same model outputs and same input context
> - Gemini 2.5 Pro and GPT-5 often disagreed on absolute scores and relative rankings.
> - Some models changed rank by 2-3 positions depending on the judge used.
> - GLM 4.5V 106B was particularly unstable as a judge.
> This variability undermines reproducibility.
>
> **(2) LLM judges produced inconsistent scores across runs**
> For proprietary reasoning models like GPT-5, `temperature=0` is not available, this means repeated evaluations of the same answers do not yield deterministic scores, and ranking variation could appear across repeated trials.
> This introduces noise that is unsuitable for benchmarking, especially when differences between strong models are often within 2-5 percentage points.
>
> **(3) Judge bias interacts with the model under evaluation**
> We observed some systematic patterns. Gemini tends to rate Gemini-style outputs more favorably. GPT-5 tends to rate GPT-family outputs higher, even when judged on identical content. This well-known "judge-model similarity bias" is documented in MT Bench [1], AlpacaEval [2] and other LLM-judge evaluations.
>
> [1] Zheng et al. Judging LLM-as-a-Judge with MT-Bench and Chatbot Arena.
>
> [2] Dubois et al. Length-Controlled AlpacaEval: A Simple Way to Debias Automatic Evaluators.

---

> > ### Author Response · Authors · 2025-11-20
> >
> > # Part 2/2
> >
> > ## Continuation of Weakness 1
> > As a conclusion, our additional open-ended evaluation shows promise for qualitative error analysis but also reveals judge inconsistency, judge-model bias, run-to-run variability and difficulty in replicating results.
> > ### Reasoning for Choice of MCQ
> > In comparison to open-ended evaluation, MCQ remains more reproducible. It provides deterministic scoring, controlled distractor difficulty, identifcal evaluation conditions for all models and no dependence on third-party proprietary judges.
> > Furthermore, our JSON-based debiasing removes the dominant shortcut patterns and brings model behavior closer to human reasoning (demonstrated via our user study).
> >
> > Therefore, we believe the MCQ format remains the more reliable and reproducible framework for PRISMM-Bench. We will include the open-ended results and insights in the camera-ready version to facilitate future exploration of hybrid evaluation protocols.
> >
> > ## Weakness 2: Scope and Scale of the Dataset
> > We appreciate the reviewer's concern and have taken it seriously. In the short rebuttal window, we invested substantial effort to assess the scalability and robustness of our benchmark.
> >
> > ### Expanded Dataset and Re-evaluation
> > To test whether our conclusions hold beyond the original set of 262 instances, we extended our pipeline to a new source: ICLR 2024, and collected and human-verified an additional 122 inconsistencies. We then re-ran all 21 model evaluations on this expanded benchmark.
> > The core performance trends and model rankings remained stable, with the only minor change being GPT-5 (high-reasoning) narrowly overtaking Gemini 2.5 Pro. This stabilitiy demonstrates that our original benchmark size is sufficient to capture the difficulty of the task, and that our data collection pipeline scales effectively to new papers.
> >
> > **We have already updated the paper on the OpenReview page with the expanded dataset statistics and the full set of re-evaluation results.**
> >
> > ### Justification of Review Source
> > Our reliance on ICLR is driven by practical constraints of open peer-review availability.
> > - ICLR is the only major venue that publicly releases full reviews for withdrawn papers, which is essential for recovering inconsistencies that still exist in the manuscript.
> > - NeurIPS and ICML do not guarantee public reviews for rejected or withdrawn submissions, making it impossible to systematically collect reviewer-identified inconsistencies at scale.
> > - We also focus specfically on rejected or withdrawn papers without a rebuttal, because reviewer-flagged inconsistencies in accepted papers are typically corrected in the camera-ready version, breaking the alignment between the flagged issues and the available PDF.
> >
> > ### Domain Coverage and Future Expansion
> > We restrict our current benchmark to AI/ML domain because human verifiation requires domain expertise to interpret subtle technical inconsistencies. Extending to other scientific fields necessitates recruiting experts from those communities, which is feasible but beyond the scope of the initial release.
> >
> > We fully agree that the broader domain coverage is important. We are already exploring expansion to other open-review ecosystem, such as F1000Reseasrch [1], which uses a transparent post-publication peer-review model and covers biomedical disciplines. We explicitly ackonwledge this as a limitation and regard our PRISMM-Bench as a first step towards broader, peer-review-grounded multimodal evaluation.
> >
> > [1] https://f1000research.com

---

> ### Author Response · Authors · 2025-11-27
> **Gentle Reminder**
>
> Dear Reviewer stE4,
>
> We would like to kindly draw your attention to our detailed rebuttal and the additional experiments we conducted in response to your comments. We have aimed to thoroughly address each of your concerns, and the revisions are marked in the updated manuscript.
> If you have a moment to review our responses, we would greatly appreciate your feedback. If our clarifications resolve your concerns, we kindly ask you to consider updating your evaluation. We are, of course, happy to clarify anything further before the discussion period closes on December 3.
>
> Thank you again for your time and constructive comments.

---

### Author Response · Authors · 2025-11-20

Dear Area Chair and Reviewers,

We sincerely thank the reviewers for their constructive feedback. We are glad to see they found our task novel and of practical value (**CmSk**), recognizing the high originality and ecological validity of grounding benchmark in real peer reviews (**stE4, KPqE, gBzJ**). We are also pleased that reviewers found our data collection pipeline reasonable and robust (**stE4, gBzJ**), our experiments comprehensive (**stE4, gBzJ**), and our JSON-based debiasing method insightful, innovative and efficient at mitigating shortcuts (**CmSk, KPqE, gBzJ**).

To address concerns regarding benchmark scale, domain coverage, and evaluation methodology, we conducted several new experiments during the rebuttal period:

- **Benchmark Expansion.** We applied our pipeline to ICLR 2024 and added 122 new validated inconsistencies. Re-running all 21 model evaluations on the expanded set showed stable rankings and trends, confirming that our original dataset size was already statistically robust. We updated the manuscript accordingly (changes marked in green).
- **MCQ vs. LLM-as-a-Judge.** We directly compared open-ended evaluation (with three LLM judges) against our MCQ protocol. The open-ended setting suffered from large cross-judge variance, nondeterministic scores, and judge–model similarity bias, making it unsuitable for benchmarking. In contrast, our JSON-based MCQ format remained fully deterministic and reproducible, and held up under adversarial distractor stress tests.
- **Inter-Annotator Agreement.** We performed a new IAA study, obtaining Cohen’s κ = 0.86 (Almost Perfect) for inconsistency detection and Substantial Agreement for category labels (which are not used in evaluation). This confirms the objectivity and consistency of our annotation protocol.
- **Human–Model Gap Robustness.** A bootstrap simulation showed that the user-study subset was somewhat easier for two models. This means our reported human–model gap is conservative, and would be even larger on a representative subset.

We appreciate your guidance, which has substantially strengthened the paper. We hope the new experiments and revisions satisfactorily address your concerns, and we kindly ask you to consider updating your rating in light of these additions. We are happy to provide any further clarification during the discussion period.

---

### Meta-Review · Area_Chair_3VLK · 2026-01-02

**Summary:**

This paper proposes a benchmark named PRISMM-Bench on real reviewer-flagged multimodal inconsistencies in scientific papers. The reviewers believe the topic is quite interesting and has potential practical value. All four reviewers gave the "borderline acceptance" rating before rebuttal, which shows that the quality of the paper is quite solid. The reviewers had some concerns about open-ended evaluation, and the authors have resolved the issue with more experiments during the rebuttal.

**Reviewer Concerns:**

See above.

**Reviewer Scores:**

I believe that the reviewers will maintain positive scores.

---

### Decision · Program_Chairs · 2026-01-26

Accept (Poster)